# Decoding the genome of *Brainea insignis* reveals insights into fern evolution and conservation

Zengqiang Xia [1,2,3], Lei Duan[1,3], Yuhan Fang[1,3], Yan Jiang[1,3], Hongfeng Chen[1,3], Yuehong Yan[2,3], Aihua Wang[4], Zixiang Li[1,3], Ziyue Liu [1,3], Guohua Zhao[5], Hui Shen[2,3], Yves Van de Peer [6,7,8,9] ✉, Ming Kang [1,3] ✉ & Faguo Wang[1,3] ✉

Ferns are an ancient lineage of vascular plants, yet limited genomic resources constrain both evolutionary and conservation inference. Here, we generate a chromosome-level genome assembly for the endangered cycad fern *Brainea insignis* (8.62 Gb), the sole species in its genus within eupolypods II, and integrate comparative and population genomics to resolve its evolutionary history and vulnerability. The genome retains the ancient whole-genome duplication shared by leptosporangiate ferns; however, its exceptional size is driven primarily by recent repeat accumulation and further shaped by lineage-specific evolutionary signatures linked to functional specialization. Resequencing across the range identifies three geographically and environmentally structured lineages shaped by Quaternary refugia, limited postglacial expansion and localized admixture. Recently reduced populations show pronounced genomic erosion, including inbreeding and elevated genetic load, due to insufficient time for purging. We detect climate-associated local adaptation and project substantial future genetic offsets, with southwestern Indochina populations at highest risk. Our results expand fern genomics and support spatially tailored conservation strategies that maintains habitat connectivity and promotes adaptive gene flow.

Ferns are among the earliest lineages of vascular plants[1] and encompass over 13,000 species distributed across diverse ecological niches worldwide[2]. These species exhibit striking morphological and ecological variation—from small, ground-hugging plants to towering tree ferns—and thrive in habitats ranging from tropical rainforests to arid or temperate zones[3,4]. As the sister lineage to seed plants, ferns provide critical insights into macroevolution transitions in plant anatomy, life history, and reproduction[1,5]. Moreover, their high spore production and effective wind dispersal[6] make many fern species ideal models for investigating gene flow, dispersal

[1]Guangdong Provincial Key Laboratory of Applied Botany, State Key Laboratory of Plant Diversity and Specialty Crops, South China Botanical Garden, Chinese Academy of Sciences, Guangzhou, China. [2]Eastern China Conservation Centre for Wild Endangered Plant Resources, Shanghai Chenshan Botanical Garden, Shanghai, China. [3]University of Chinese Academy of Sciences, Beijing, China. [4]Key Laboratory of Environment Change and Resources Use in Beibu Gulf, Ministry of Education, and Guangxi Key Laboratory of Earth Surface Processes and Intelligent Simulation, Nanning Normal University, Nanning, China. [5]Shenzhen Key Laboratory of Southern Subtropical Plant Diversity, Fairy Lake Botanical Garden, Shenzhen & Chinese Academy of Sciences, Shenzhen, Guangdong, China. [6]Department of Plant Biotechnology and Bioinformatics, Ghent University, Ghent, Belgium. [7]VIB-UGent Center for Plant Systems Biology, Ghent, Belgium. [8]Department of Biochemistry, Genetics and Microbiology, University of Pretoria, Pretoria, South Africa. [9]College of Horticulture, Academy for Advanced Interdisciplinary Studies, Nanjing Agricultural University, Nanjing, China. ✉e-mail: yvpee@psb.vib-ugent.be; mingkang@scbg.ac.cn; wangfg@scbg.ac.cn

patterns, and local adaptation across broad spatial and ecological gradients.

Despite significant progress in seed plant genomics, which has illuminated the complex genomes of many economically and ecologically important species[7–9], fern genomics remains comparatively understudied. This disparity is largely attributable to the large genome sizes and complex ploidy levels characteristic of many ferns[10,11]. For instance, ferns commonly possess large genomes and intricate chromosome structures, as exemplified by *Tmesipteris oblanceolata*, which has a record-breaking 160-Gb genome[12]. Additionally, most population genetic theories and analytical tools are tailored to diploid species[13,14], limiting their applicability to ferns, which often deviate from these genetic and genomic norms[10,15]. As a result, fewer than ten fern species currently have fully assembled genomes, leaving large gaps in our understanding of fern diversification, adaptation, and evolutionary trajectories.

Within ferns, eupolypods—subdivided into eupolypods I (Polypodiineae) and eupolypods II (Aspleniineae)—represent roughly two-thirds of extant fern diversity[16]. Despite their ecological and evolutionary significance, no high-quality reference genome has been established for this clade. *Brainea insignis*, commonly known as the cycad fern, belongs to eupolypods II and is the sole species in its genus. This monotypic genus is found in tropical Asia and bears considerable ornamental and medicinal importance[17–19]. It has been listed as a nationally protected species in China since 1999[20] and identified as a priority protected species in India[21]. In China, *B. insignis* is highly sensitive to environmental disruption and human activities, which have led to precipitous population declines[17]. Given the species' dual importance to conservation and evolutionary studies, understanding its genomic makeup and evolutionary resilience is critical for guiding the conservation of both the species itself and the biodiverse habitats it sustains[22].

In this study, we present chromosome-level genome assembly for *B. insignis*, providing an essential genomic resource for the eupolypods II clade. By resequencing 94 individuals from multiple populations, we investigate patterns of genetic diversity, population structure, local adaptation, and the demographic history of this endangered species. Building upon these genomic data, we further estimate genetic offset under future climate scenarios to assess the risks of climate-induced maladaptation. Our results bridge a critical gap in fern genomics, offering fresh insights into the mechanisms underpinning fern genome evolution and informing evidence-based conservation strategies. By addressing key questions regarding the evolutionary potential of *B. insignis*, our work not only contributes to the broader understanding of fern diversification but also highlights the importance of genomics-based strategies for preserving biodiversity in a rapidly changing world.

## Results and discussion

### Chromosome-scale genome assembly and annotation

To guide our sequencing strategy, we first determined the genome size and chromosome complement of *B. insignis*. Four independent flow cytometry analyses estimated the genome size at 8.71 Gb (Supplementary Fig. 1), while cytological examination revealed a somatic chromosome number of $2n = 68$ (Supplementary Fig. 2). A subsequent genome survey suggested a genome size of approximately 8.40 Gb, with low heterozygosity (0.28%), a high repeat content (89.23%), and diploidy, as indicated by k-mer frequency analyses (Supplementary Fig. 3). Building on these findings, we generated a deep-sequencing dataset to account for the large genome size and the abundance of repeats. We obtained a total of 359.99 Gb (41.76× coverage) of PacBio HiFi reads and 1404.44 Gb (162.92× coverage) of Hi-C reads (Supplementary Table 1 and 2). We then assembled an 8.62 Gb genome, with a contig N50 of 4.36 Mb and a scaffold N50 of 265.61 Mb (Table 1). We anchored 99.81% of the contig length to 34 pseudochromosomes (Fig. 1, Table 1, and Supplementary Fig. 2), corresponding to the 34 chromosomes of the *B. insignis* haploid set ($n = 34$). This represents one of the largest haploid genomes with a chromosome-level assembly reported for a non-seed plant, surpassing many existing seed plant assemblies in size.

We assessed the quality of the assembled genome using multiple approaches. First, 99.66% of Illumina short reads (excluding supplemental and secondary reads) mapped to the assembly. Second, BUSCO analysis (viridiplantae_odb12 dataset; updated July 1, 2025) indicated that 97.4% of the 822 conserved genes were completely recovered (Supplementary Table 3). Third, the assembly achieved a LTR Assembly Index (LAI) score of 10.34, surpassing the reference-level threshold of 10. Finally, Clipping Information for Revealing Assembly Quality (CRAQ) analysis showed strong structural accuracy, with a regional assembly quality (R-AQI) of 96.17 and a structural assembly quality (S-AQI) of 98.73. These results collectively underscore the high contiguity, consistency, and completeness of our *B. insignis* assembly.

Repetitive elements, particularly transposable elements (TEs), account for a large fraction of fern genomes[23,24], and *B. insignis* is no exception. Overall, 7.06 Gb (81.77%) of the genome comprises repeat sequences (Supplementary Table 4), aligning with the genome survey estimate of 89.23% repetition (Supplementary Fig. 3). Long terminal repeat retrotransposons (LTR-RTs) dominate this repeat landscape, accounting for 47.99% of the genome, followed by DNA transposons (DNATs) at 24.49%. The remainder includes non-LTR retrotransposons (e.g., LINEs and SINEs) and a small fraction (2.8%) of unclassified repeats (Supplementary Table 4).

We predicted a total of 43,573 protein-coding genes within the assembled genome (Supplementary Fig. 4a; Supplementary Table 5), with 89.78% of these genes having functional annotations in six major databases (Supplementary Fig. 4b and Supplementary Table 6). The structural characteristics of these genes are broadly consistent with those reported for other fern species (Supplementary Fig. 5): on average, each gene is 15,096.24 bp in length, contains 3.84 exons with a mean exon length of 282.76 bp, and has introns spanning 4,939.39 bp (Supplementary Table 5). Additional BUSCO analysis showed that 692 (84.2% of 822) viridiplantae_odb12 genes were present as complete genes in the annotation (Supplementary Table 3). Collectively, these metrics illustrate the robust quality of our structural and functional gene predictions for *B. insignis*.

### Genomic structural features and comparative genomics

We first examined the structural attributes of the *B. insignis* genome by profiling gene density, GC content, and the distribution of transposable elements (TEs), particularly long terminal repeat retrotransposons (LTR-RTs) from the Gypsy and Copia families (Fig. 1). Both

**Table 1 | Genome assembly statistics of *B. insignis***

| Contig level | Value |
| --- | --- |
| Assembly size (Gb) | 8.78 |
| Number of contig | 5,791 |
| Contig N50 (Mb) | 4.36 |
| N50 contig number | 626 |
| GC content | 41.59% |
| PE reads mapping rate | 99.66% |
| **Chromosome level** | **Value** |
| Total length of chromosomes (Gb) | 8.62 |
| Scaffold N50 (Mb) | 265.41 |
| Anchor ratio | 99.81% |
| BUSCO completeness | 97.4% |
| LAI | 10.34 |
| R-AQI/S-AQI | 96.17/98.73 |

LAI: LTR assembly index; R-AQI: regional assembly quality; S-AQI: structural assembly quality.

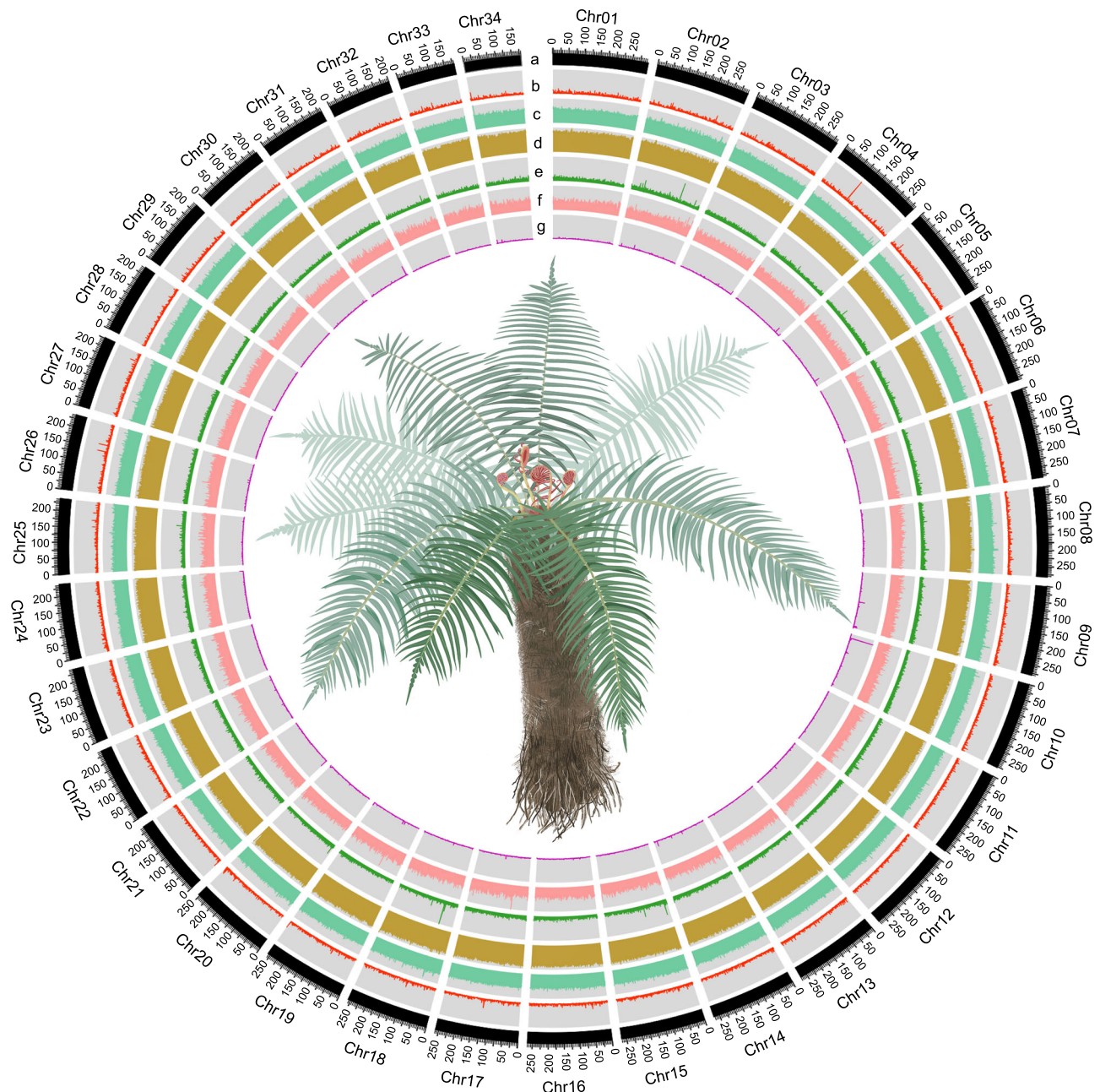

**Fig. 1 | Genome features and morphological illustration of *B. insignis*.** The genomic landscape of the 34 *B. insignis* pseudo-chromosomes is shown in non-overlapping 600-kb windows. Panels **a**–**g** correspond to tracks from the outer to inner rings of the circos plot. **a** chromosome length, **b** gene density, **c** GC content, **d** TE coverage, **e** LTR-Copia coverage, **f** LTR-Gypsy coverage, and (**g**) tandem repeats density.

genes and repeats showed a relatively uniform distribution across the genome, in contrast to the more localized patterns typically observed in seed plants (e.g., *Glycine max* and *Miscanthus floridulus*), where gene density generally increases near chromosome termini[25,26]. Similar homogeneity in genomic architecture has been reported for other homosporous fern genomes, including *Ceratopteris richardii*[27], *Alsophila spinulosa*[28], and *Adiantum capillus-veneris*[29], as well as in select lycophytes (*Isoetes taiwanensis*[30], *Huperzia asiatica*[31], *Diphasiastrum complanatum*[31]). Although ferns and seed plants share a common ancestor, these findings suggest that lycophytes and ferns may exhibit more similar genomic structural patterns than seed plants. Accordingly, the differences in gene density and repeat coverage between seed-free and seed-bearing vascular plants may be more nuanced than previously assumed[24,29].

Whole-genome duplication (WGD) is a pivotal force shaping genome architecture and driving evolutionary innovation[15]. Synonymous substitution rates ($K_s$) distributions are constructed by calculating the $K_s$ between pairs of homologous genes. To explore the role of WGD in *B. insignis*, we employed a rate-corrected $K_s$ distribution[32], which allowed the rescaled orthologous divergence times to be comparable with the paranome $K_s$ distribution of the focal species. A prominent $K_s$ peak at 1.77 suggests an ancient WGD event predating the divergence of *Ceratopteris richardii*, *Adiantum capillus-veneris*, *Alsophila spinulosa*, and *Marsilea vestita* (Fig. 2a). Further analysis of paralogous $K_s$ distributions and collinearity corroborated this ancient WGD[33], pinpointing a consistent peak (~1.7) and revealing no subsequent, lineage-specific WGDs (Supplementary Figs. 6 and 7). These findings align with genome analyses of *A. capillus-veneris*, which also

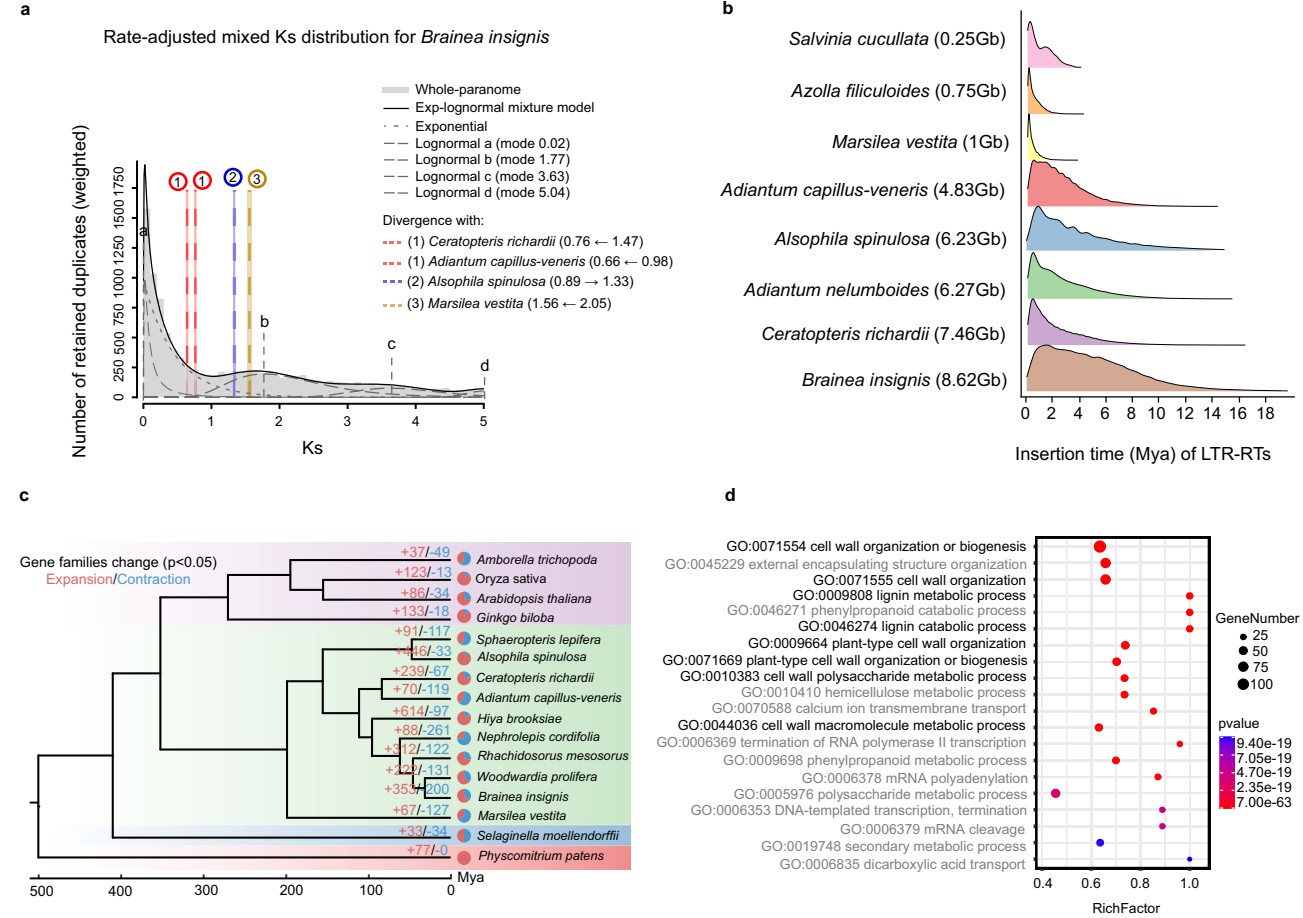

**Fig. 2 | Comparative genomics analyses. a** Rate-adjusted mixed $K_s$ distribution of *B. insignis*. The whole paranome $K_s$ distributions are overlaid with rate-adjusted divergence events (speciation events) in colored vertical lines and boxes. As described in Chen et al.[80], the overall mixture model, represented as the dark solid KDE curve, is made up of an exponential component (dotted grey curve) and optimized log-normal components (dashed grey curves). The log-normal components are labelled with letters. The horizontal arrows indicate the $K_s$ shifts resulting from substitution rate adjustment. **b** Estimated LTR-RT insertion time distributions for ferns, calculated with lineage-specific substitution rates. **c** Significant gene family expansion and contraction among 16 plant species, including one bryophyte

(outgroup), one lycophyte, four seed plants, and ten ferns. Red and blue numbers above branches denote expansion and contraction events, respectively. Statistical significance was assessed using CAFE's birth–death model (two-sided), and *P* values were obtained directly from CAFE without additional multiple-comparison correction. **d** GO enrichment of the 353 significantly expanded gene families in *B. insignis*. Enrichment was tested using a one-sided hypergeometric test, and *P* values were corrected using the Benjamini–Hochberg FDR method (significant at corrected *P* ≤ 0.05). RichFactor denotes the ratio of expanded to total annotated genes per GO term; bubble size shows gene number and colour indicates the adjusted *P* value.

exhibit only this ancient WGD shared among core leptosporangiate ferns[29].

Interestingly, the $K_s$ peak value for *B. insignis* (-1.7) is lower than that of *A. capillus-veneris* (-2.15), implying a comparatively slower synonymous substitution rate in *B. insignis*. To investigate this further, we quantified synonymous (dS) and nonsynonymous (dN) substitution rates in 16 fern species. *B. insignis* and its close relatives (*Woodwardia prolifera*) showed significantly lower dS and dN values than many other ferns (Supplementary Fig. 8). Indeed, relative rate tests indicate that *B. insignis* exhibits a significantly slower evolutionary rate compared to most other core leptosporangiate ferns (Supplementary Table 7), even slower than certain tree fern species (e.g., *Alsophila spinulosa*). Whole-genome collinearity with the tree fern *A. spinulosa* revealed extensive one-to-one syntenic blocks in *B. insignis* (Supplementary Fig. 9). Despite over 200 million years of divergence (TimeTree5), the breadth of this conserved collinearity indicates that the *B. insignis* genome has evolved comparatively slowly, at least in terms of genomic architecture. Although *B. insignis* exhibits a higher ω (dN/dS) ratio than most core leptosporangiate ferns (Supplementary Fig. 10), this may reflect relaxed selective pressures–possibly owing to a stable habitat or small effective population size–rather than accelerated protein

evolution. In turn, this slow evolutionary rate may have bolstered genomic stability, enabling *B. insignis* to persist as the sole species in its genus, although it could also limit its adaptive potential in the face of rapid environmental change.

In addition to WGD, variations in LTR-RT content strongly influence genome size evolution in many plant lineages[24,34,35]. However, their specific impact on fern genome expansion remains poorly understood, largely due to limited genomic data. To address this gap, we compared LTR-RT composition and insertion times in eight fern genomes. Our findings indicate that larger fern genomes tend to harbor a higher proportion of LTR-RTs with earlier insertion time (Fig. 2b; Supplementary Fig. 11). While these observations suggest a positive correlation between LTR-RT activity and genome size expansion in ferns, the limited number of available fern genomes hampers broad statistical inferences. Consequently, more extensive sampling and high-quality assemblies are needed to better understand how LTR-RT dynamics have shaped the remarkably diverse genome architectures across ferns.

To further investigate gene family evolution in the *B. insignis* genome, we constructed a high-confidence phylogeny for 16 species using 103 single-copy gene families (Fig. 2c). We identified

353 significantly expanded and 200 significantly contracted gene families on the branch leading to *B. insignis*, with the expanded families enriched in biological processes related to cell wall organization and lignin metabolism ("plant-type cell wall organization", "lignin metabolic process" and "lignin catabolic process"; Fig. 2d). Gene families involved in monolignol biosynthetic pathway exhibit $K_a/K_s < 1$ in *B. insignis* and two other tree ferns (Supplementary Fig. 12), and microsynteny analyses further reveal that a subset of these lignin-related genes is highly conserved across species (Supplementary Fig. 13). These findings suggest that the ancestral lineage of *B. insignis* evolved specialized lignin-related traits—such as robust, lignified structures—that have been retained in the modern species, while key metabolic pathways remained long-term functional stability. Interestingly, a recent study showed that *Stenochlaena palustris*, a member of the same family, possesses remarkable lignin architectures[36], indicating that S-lignin production evolved independently in ferns. Although lignin composition in *B. insignis* remains to be characterized, the observed expansion and synteny suggest lineage-specific elaboration of lignin-related capacities. Together, these observations are consistent with lineage-specific diversification of lignin pathways in ferns.

Overall, our findings demonstrated that an ancient WGD, repetitive-element dynamics, and gene family expansions have collectively sculpted the genome of *B. insignis*. Although *B. insignis* shares a deep polyploidization event with other core leptosporangiate ferns, it exhibits a slower evolutionary rate and a larger genome, underscoring the complexity of diploidization processes and the importance of repetitive elements in shaping fern genome diversity.

## Population structure and demographic history

We resequenced 94 *B. insignis* individuals from 29 geographic locations to an average depth of 21.81×, generating 17.79 Tb of raw data (Fig. 3a; and Supplementary Fig. 14 and Supplementary Data 1). After mapping paired-end reads to the *B. insignis* reference genome and applying stringent filtering criteria, we identified 75,060,153 high-quality SNPs and 9,414,109 core variants defined as a high-confidence SNPs set retained after stringent quality control (Supplementary Fig. 15). Dataset 1-3 were derived from these variants and showed no evidence of chromosomal bias (Supplementary Fig. 16).

To elucidate the genetic structure of *B. insignis*, we applied several methods, including ADMIXTURE, principal components analysis (PCA), neighbor-joining (NJ) trees, and chloroplast haplotype networks. ADMIXTURE identified two primary clusters (K = 2), differentiating Yunnan populations (YN lineage) from those in southern China (SC lineage). The optimal number of clusters was determined to be three (K = 3), revealing a distinct lineage from Vietnam (VN lineage) and two apparent admixture zones (Admixture1 and Admixture2) (Fig. 3b; and Supplementary Fig. 17). Additional analyses corroborated this tri-lineage pattern: (1) the chloroplast haplotype network revealed clear distinctions among the YN, VN, and SC lineages, with apparent admixture haplotypes (Supplementary Fig. 18); and (2) PCA and NJ trees confirmed genetic separations corresponding to these three main lineages while highlighting intermediate signatures in the admixture zones (Supplementary Fig. 19). Together, these results show strong isolation by distance (IBD) and isolation by environment (IBE), suggesting that both geographic distance and environmental heterogeneity jointly influence the spatial distribution of genetic variation. This pattern is consistent with multiple glacial refugia during the Quaternary, followed by post-glacial expansions and secondary contact among lineages.

We reconstructed the demographic trajectories of each lineage using the pairwise sequentially Markovian coalescent (PSMC) method (Fig. 3c). All lineages exhibited similar initial trends: an increase in effective population size (*Ne*) peaking around 2 million years ago (Mya), followed by a protracted decline until approximately 300 thousand years ago (Kya). A subsequent expansion culminated in a sharp bottleneck around 200 Kya, coinciding with the onset of Marine Isotope Stage 6 (MIS6) glaciation[37,38]. These observations mirror the glacial-contraction and interglacial-expansion cycles reported for other East Asian relict species, including *Ginkgo biloba*[39], *Cercidiphyllum japonicum*[40], and *Davidia involucrata*[41]. After the MIS6 bottleneck, the YN lineage diverged from the common ancestor around 100 Kya, whereas the VN and SC lineages separated around 40 Kya. Each lineage subsequently maintained relatively small and stable *Ne* values up to the present (Fig. 3c; and Supplementary Fig. 20).

To complement the PSMC analyses, we performed coalescent-based simulations for each lineage using fastsimcoal2 and identified the most likely speciation model among five initial scenarios without introgression (Supplementary Fig. 21 and 22; Supplementary Table 8 and 9). After determining the best-fitting model, we tested nine additional models incorporating gene flow (Supplementary Fig. 23; and Supplementary Table 10). The optimal model corroborated the PSMC-derived divergence times (Fig. 3d) and provided a robust framework for understanding the evolutionary history of *B. insignis*, further supporting its status as a relict fern species[17,20]. Our simulations also revealed variations in gene flow across different divergence phases. Substantial sharing of identical-by-descent haplotypes between the YN/SC lineages and the admixture zones (Admixture1 and Admixture2) implies relatively recent inter-lineage gene flow (Fig. 3e). Furthermore, TreeMix analysis, with an optimal migration edge of 1, indicates recent gene flow from YN to VN (Supplementary Fig. 24), underscoring ongoing genetic exchange and secondary contact among geographically proximate populations.

## Genomic effects of recent population decline on inbreeding and genetic load

Although effects of population declines are well-documented in many seed plants and animal lineages, the genomic consequences of population decline in ferns remain largely unexplored[42]. To address this gap, we analyzed genetic diversity, inbreeding, and genetic load in *B. insignis*. We hypothesized that the consistently low effective population size (*Ne*) since the last bottleneck, compounded by severe declines in recent decades, has led to further losses of genetic diversity. Indeed, genome-wide heterozygosity in the three lineages (YN, VN, and SC) remains low, averaging 0.08, 0.12, and 0.10, respectively (Fig. 4a). Accordingly, the YN lineage exhibits the lowest nucleotide diversity (mean $\pi = 1.043 \times 10^{-3}$), followed by VN ($1.126 \times 10^{-3}$) and SC ($1.379 \times 10^{-3}$) (Supplementary Fig. 25). YN also exhibits slower linkage disequilibrium (LD) decay, which is consistent with its lowest level of genetic diversity. Genetic divergence ($F_{ST}$) values (0.116–0.320) correlate with the geographic distances between the lineages, while positive Tajima's *D* values corroborate the demographic bottlenecks inferred from our historical reconstructions.

To quantify inbreeding, we identified runs of homozygosity (ROH) in each lineage. All three lineages show numerous long ROHs (>100 kb), with the fraction of the genome in ROH ($F_{ROH}$) averaging 0.51 (YN), 0.50 (VN), and 0.43 (SC) (Fig. 4b). Furthermore, a substantial proportion of ROHs exceed 1 Mb (Fig. 4c), indicating pronounced inbreeding. We observed a strong negative correlation between individual heterozygosity and $F_{ROH}$ ($R^2 = 0.77$, $P < 0.001$), confirming that high levels of inbreeding are reducing genome-wide diversity (Fig. 4d).

Next, we assessed genetic load by examining allelic states at loss-of-function (LOF) and missense (deleterious) variants. The ratio of nonsynonymous to synonymous ($\pi_0/\pi_4$) diversity was relatively high, ranging from 0.431 to 0.503 (Supplementary Table 11), suggesting that purifying selection on 0-fold sites has been weak or relaxed in *B. insignis*. To put our estimates in a broader context, we conducted a comparative analysis of $\pi_0/\pi_4$ ratios across more than 40 plant species, spanning ferns and seed plants (Supplementary Fig. 26). The endangered ferns (*B. insignis* and *Alsophila* spp.) exhibit markedly elevated

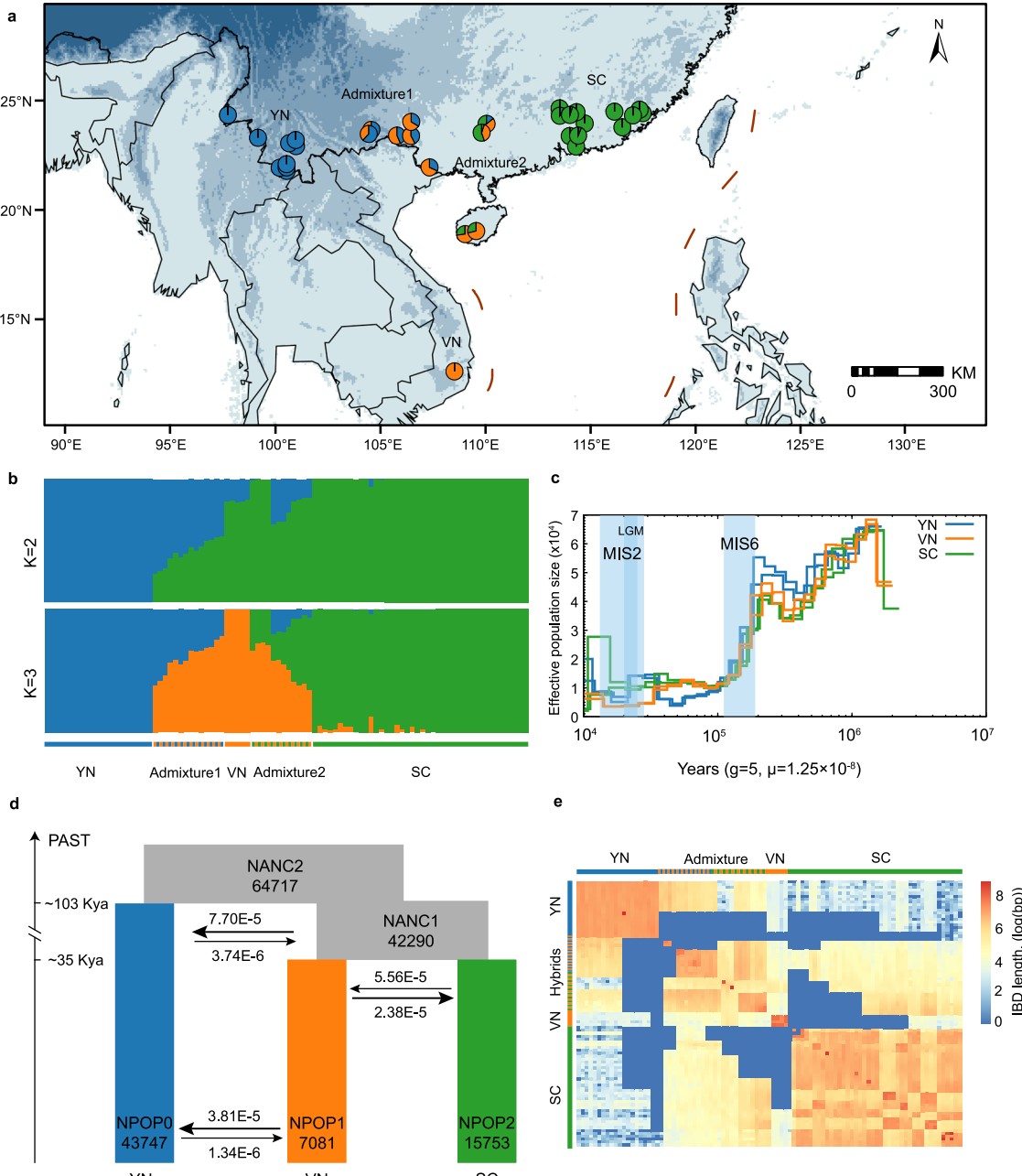

**Fig. 3 | Genetic structure and demographic history of *B. insignis*. a** Map showing sampling locations and overall genetic structure. **b** Population structure of 94 individuals based on ADMIXTURE analysis at *K* = 2 and 3. **c** Demographic history of the three lineages of *B. insignis*, with the three glacial events (MIS2, LGM, and MIS6) highlighted in light blue. **d** Schematic of the best demographic scenario inferred with fastsimcoal2. Numbers at the bottom indicate the estimated *Ne* of each lineage; and the bidirectional arrow shows the direction of gene flow. **e** Heatmap illustrating haplotype sharing among individuals, where colors represent the total length of identity-by-descent (IBD) blocks for each pairwise comparison.

$\pi_0/\pi_4$ ratios relative to most seed plants, with the notable exception of *Acer yangbiense*, a species characterized by an extremely small population size. These results support the expectation that small population sizes weaken purifying selection and inflate genetic load across disparate plant lineages. Although absolute values may vary with sampling and analytical pipelines, the qualitative pattern is robust. Deleterious (DEL) and LOF mutations in the homozygous state—an indicator of genetic load[43,44]—are most frequent in the YN lineage (Fig. 4e, f). Moreover, applying a Grantham Score threshold of 150 further underscores the elevated fraction of deleterious missense variants in YN lineage (Supplementary Fig. 27). The frequency of homozygous DEL and LOF alleles correlates with $F_{ROH}$ (Fig. 4g, h). We also observed a higher density of LOF variants within ROH segments

than outside them (Fig. 4i), highlighting the role of inbreeding in exacerbating genetic load. Functional annotations link LOF-afflicted genes to stress resistance and DNA repair (Supplementary Data 2)—processes vital for organismal survival—implying that accumulated deleterious mutations may diminish population fitness.

Collectively, our findings demonstrate that *B. insignis* harbors extremely low genetic diversity in all three lineages and exhibits pronounced inbreeding and genetic load, particularly in the YN lineage. This pattern contrasts with long-term small populations (e.g., *Alsophila spinulosa*)[42], where signals of purging have been reported[43,45–48]. In *B. insignis*, however, a recent and severe population contraction likely limited purging opportunities, thereby facilitating the accumulation of deleterious alleles.

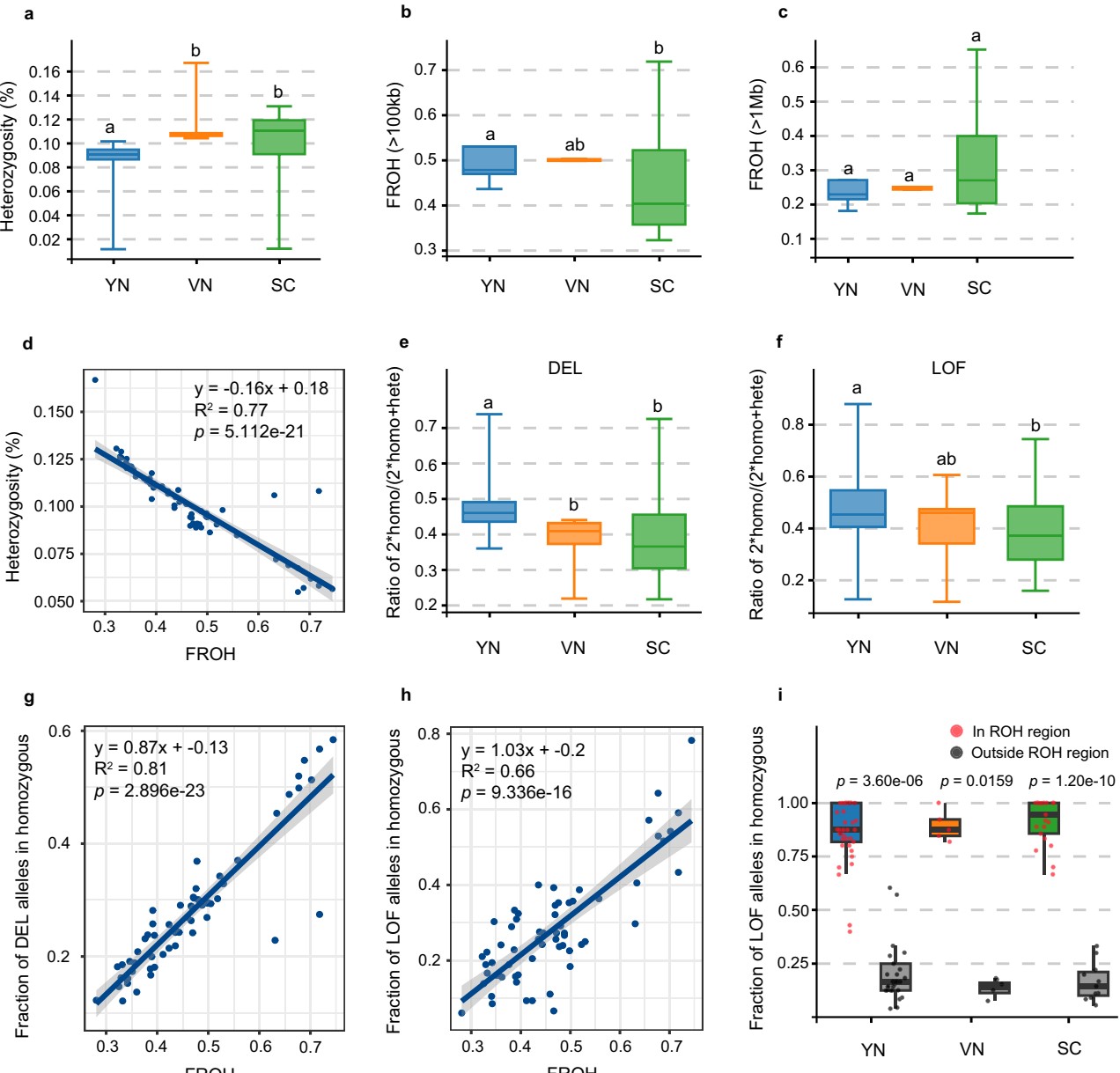

**Fig. 4 | Characterization of inbreeding and genetic loads in *B. insignis*.** Population genetic characteristics of three lineages (YN: $n = 21$, VN: $n = 5$, SC: $n = 42$ individuals). For all box plots, the box represents the 25th–75th percentiles, the center line indicates the median, and whiskers show the minimum and maximum values. Different lowercase letters (**a**, **b**) indicate significant differences between lineages. **a** Genome-wide heterozygosity estimates of three lineages. Exact $P$ values: YN vs. VN = 0.00003, YN vs. SC = 0.00063, VN vs. SC = 0.77612. **b** Inbreeding levels ($F_{ROH}$) based on medium-length segments ($>100$ kb). Exact $P$ values: YN vs. VN = 0.85, YN vs. SC = 0.014, VN vs. SC = 0.7. **c** Inbreeding levels ($F_{ROH}$) based on long segments ($>1$ Mb). Exact $P$ values: YN vs. VN = 0.71, YN vs. SC = 0.45, VN vs. SC = 0.27. **d** Regression of genome-wide heterozygosity against $F_{ROH}$. **e** Genetic load measured by the ratio of homozygous DEL (2*homo) to the sum of homozygous and heterozygous deleterious (homo + hete). Exact $P$ values: YN vs. VN = 0.0152, YN

vs. SC = 0.0015, VN vs. SC = 0.9854. **f** Genetic load measured by the ratio of homozygous LOF (2*homo) to the sum of homozygous and heterozygous LOF (homo + hete). Exact $P$ values: YN vs. VN = 0.52, YN vs. SC = 0.04, VN vs. SC = 0.72. **g** Regression of the fraction of homozygous DEL alleles in against $F_{ROH}$. **h** Regression of the fraction of homozygous LOF alleles against $F_{ROH}$. **i** Comparison of homozygous LOF variant sites within ROH regions vs. outside ROH regions. The dot box plot shows the number of homozygous LOF variants in ROH and non-ROH regions. Statistical notes: For **a**–**c**, **e**, **f**, and **i** significant differences between lineages were assessed using two-sided Wilcoxon rank-sum tests. For the regression analyses (**d**, **g**, and **h**), the significance of the regression coefficients was evaluated using two-sided tests, and the shaded areas represent the 95% confidence intervals (CIs) of the regressions. Source data are provided as a Source Data.

Consequently, these results underscore that fern lineages shaped by different demographic histories can display sharply contrasting patterns of genetic load purging. In the case of *B. insignis*, the elevated genetic load presents serious challenges to its long-term viability, emphasizing the need for conservation measures that safeguard both habitat protection and the genetic health of the remaining populations.

## Adaptive differentiation and genetic vulnerability under future climate change

To identify genes potentially involved in lineage-specific adaptations, we conducted selective sweep analyses on the southern China (SC) and Yunnan (YN) lineages, which had large sample sizes and exhibited the highest genetic differentiation. We detected 3,846 positively selected sites (involving 116 genes) in the SC lineage and 2,299 such sites

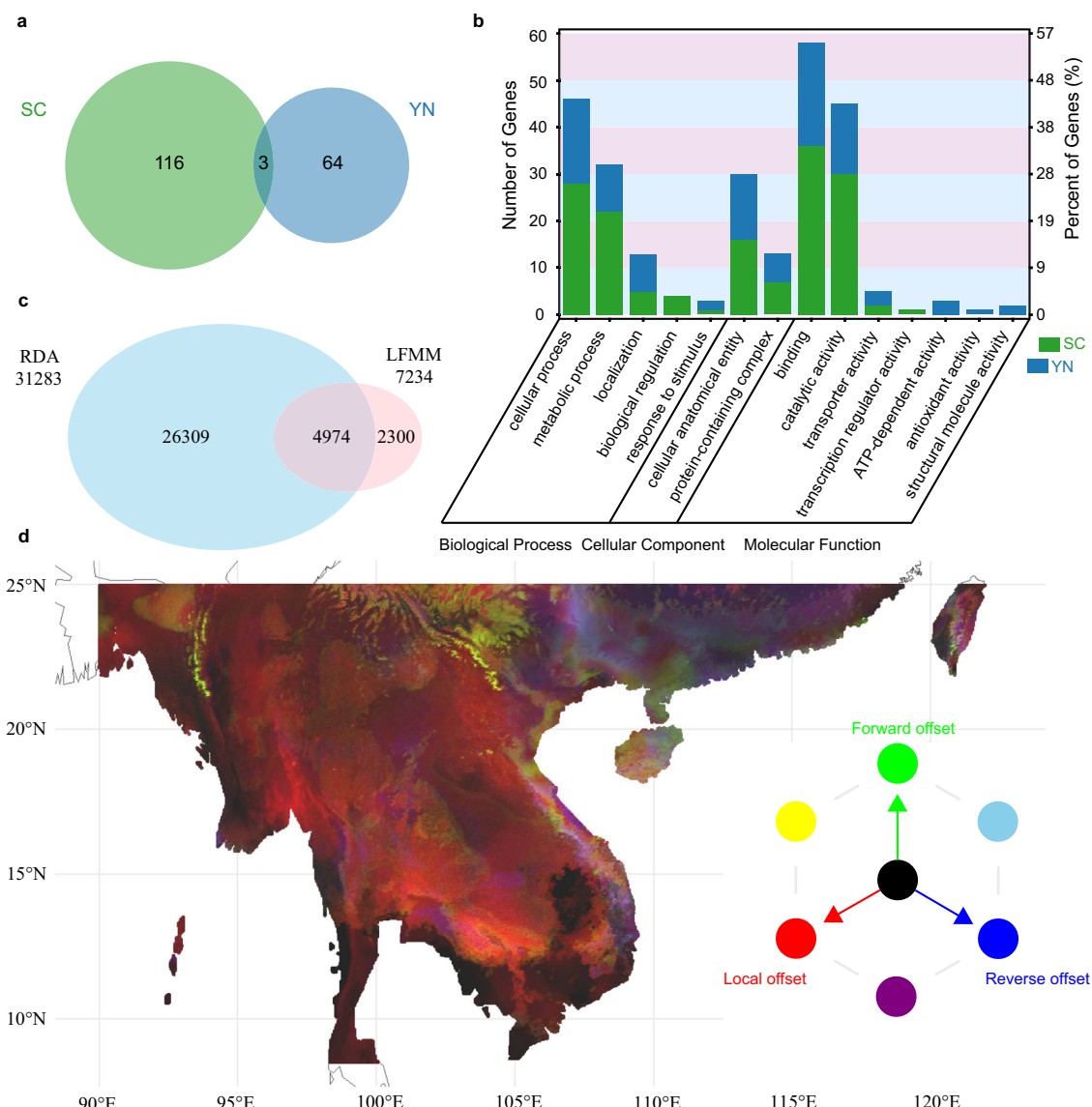

**Fig. 5 | Adaptive differentiation and genetic incompatibility analyses. a** Venn diagram of selectively swept genes identified in the two lineages. **b** GO annotation of the selected genes in each lineage. **c** Venn diagram showing overlapping loci identified by LFMM and RDA analyses. **d** RGB map of genetic offsets across the *B. insignis* range for 2081-2100 under the SSP585 scenario. Brighter cells indicate higher values for each of the three axes, whereas darker cells indicate lower values.

(encompassing 64 genes) in the YN lineage (Fig. 5a; and Supplementary Fig. 28). Only three of these genes were shared between the two lineages. Gene Ontology (GO) analysis revealed that most positively selected genes were linked to core cellular and metabolic processes, including binding and catalytic functions (Fig. 5b). Although the SC and YN lineages shared relatively few positively selected genes, they exhibited considerable overlap in functional categories, suggesting functional convergence on likely adaptive traits. Nonetheless, differences in gene counts per category and lineage-specific annotations point to diverging selection pressures that have driven adaptive differentiation.

Next, we examined the relationship between allele frequency and environmental variables in the context of local adaptation. We identified 4,974 core adaptive variants (Fig. 5c) that displayed significant isolation-by-distance (Mantel's $r = 0.4301$, $P < 0.05$) and isolation-by-environment (Mantel's $r = 0.4058$, $P < 0.05$) patterns (Supplementary Fig. 19). However, after controlling for geographic and environmental effects separately, neither relationship remained significant (partial Mantel tests: $P = 0.147$ for IBE, $P = 0.179$ for IBD), suggesting that both factors jointly shape adaptive genetic variation. Redundancy analysis

(RDA) identified three distinct genetic clusters corresponding to the YN, SC, and VN lineages, with admixture populations occupying intermediate positions (Supplementary Fig. 29). The YN lineage correlated with a high diurnal temperature range (bio2), while the SC lineage was linked to low bio2, high annual mean temperature (bio1), and high precipitation in the driest month (bio14). In contrast, the VN lineage formed its own distinct environmental cluster, and the admixture populations displayed heterogeneous environmental signatures. Collectively, these findings support local adaptation in *B. insignis*, driven by the interplay of environmental variation and geographic distance.

To evaluate the risk of genetic maladaptation under global warming, we employed a genetic offset framework[49,50]. Under various emission scenarios, higher greenhouse gas concentrations consistently produced larger offsets, signaling a greater risk of maladaptation (Supplementary Fig. 30). Populations in the southwestern Indochinese Peninsula exhibited relatively high local offsets, whereas populations outside this region showed lower offsets (Supplementary Fig. 31a, d). Forward and reverse offsets were generally very low across the species' range, suggesting that future habitats may remain suitable

(Supplementary Fig. 31b, c, e, f). However, these theoretical predictions must be interpreted in light of dispersal limitations, as *B. insignis* spores typically travel only tens to hundreds of kilometers[51]. Under high genetic offsets (> 0.15), most populations are strongly locally adapted to future climates, with only a few in the Indochinese Peninsula requiring long-distance migration (Supplementary Fig. 32a). Considering moderate offsets (> 0.05), the required migration distances increase by several thousand kilometers, with many populations needing to disperse over 5,000 km to avoid maladaptation, far exceeding their natural dispersal capacity (Supplementary Fig. 32b). Collectively, these results indicate that populations in the southwestern Indochinese Peninsula face elevated genetic vulnerability and high risk under future climate change (Fig. 5d). Polar plotting revealed no clear directional trend for potential migration (Supplementary Fig. 33), reinforcing the likelihood that range fragmentation, rather than large-scale migration, will shape the future distribution of *B. insignis*. Overall, these findings highlight not only the magnitude of genetic vulnerability but also the limited evolutionary and dispersal capacity of *B. insignis* under rapid climate change. The pronounced regional differences in offset levels suggest that conservation strategies may need to be spatially tailored, prioritizing populations in the southwestern Indochinese Peninsula. Moreover, the extreme migration distances implied by moderate offsets emphasize that natural dispersal alone is unlikely to maintain adaptive potential, raising the possibility that assisted gene flow or habitat connectivity restoration may be necessary to safeguard long-term persistence.

## Methods

### Plant materials and genome sequencing
Fresh fronds from a single *B. insignis* individual were collected from the South China National Botanical Garden in Guangzhou, China. Total genomic DNA was extracted using the CTAB method[52]. A preliminary genome survey was performed on the BGI DNBSEQ™ platform, generating 512.37 Gb of short-read data to estimate genome size, heterozygosity, and repeat content. High-molecular-weight DNA libraries (15–18 kb) were constructed and sequenced on the PacBio Sequel II/IIe platform in CCS (HiFi) mode, generating 359.99 Gb of HiFi reads with an average read length of 16.54 kb. Hi-C libraries were prepared according to established protocols[53,54]. Briefly, fresh fronds were fixed with 4% formaldehyde, digested with DpnII, and biotin-labeled DNA ligation products were sheared and used to build Illumina PE150 libraries. Five Hi-C libraries were sequenced; each amplified for 12–14 PCR cycles.

### Genome size estimation and chromosome counting
A 17-mer frequency distribution was generated using KmerFreq (v4.0 in GCE v.1.0.2), and the genome size was calculated in GCE (v.1.0.2)[55]. Flow cytometry was performed for confirmation: leaf nuclei from *B. insignis* and the internal control *Camellia sinensis var. assamica*[56] were co-stained with propidium iodide and analyzed using a BD FACScalibur flow cytometer at 488 nm excitation. The C-values were calculated based on the PI fluorescence peaks of the sample and control. For chromosome counting, root tips (~1 cm) from the sequenced individual were pretreated with p-dichlorobenzene for 5 h at room temperature, fixed in 3:1 (v/v) ethanol-to-acetic-acid solution at 4 °C, then macerated in 1 M HCl at 60 °C for 8 min. Chromosomes were stained with carbol fuchsin and observed by optical microscopy[57].

### Genome assembly and quality assessment
Genome assembly was performed with Hifiasm (v.0.19.5-r587)[58] in Hi-C mode using default parameters, incorporating both HiFi and Hi-C reads. The primary assembly was 8.78 Gb in length, consisting of 5,791 contigs (contig N50 of 4.36 Mb; Table 1). Hi-C reads were mapped to the primary assembly with BWA (v.0.7.17-r1188)[59], scaffolding was performed using Juicer (v.1.6)[60] and 3D-DNA pipeline[61], followed by manual curation using Juicebox module.

Assembly quality was evaluated by mapping short reads with BWA-MEM, completeness was assessed with BUSCO (viridiplantae_odb12 dataset)[62]. Additionally, N50 and other quality metrics were obtained using QUAST (v.5.2.0)[63]. Finally, CRAQ (v.1.0.9)[64] was used to identify regional (R-AQI) and structural (S-AQI) assembly errors.

### Genome annotation
We employed an integrated approach of homology alignment and de novo search to annotate repetitive elements. RepeatMasker (http://www.repeatmasker.org) with Dfam[65] and Repbase[66] databases was employed for homology-based detection. For ab initio prediction, a de novo transposable element (TE) library was generated using MITE-Hunter[67] and RepeatModeler (v.2.0.3)[68]. The unknown elements were classified by DeepTE[69], and a non-redundant TE library was produced via Uclust[70]. Final repeats were masked with RepeatMasker.

Protein-coding genes were predicted using ab initio, homology-based, and RNA-seq–assisted approaches (details in Supplementary Note 1). Functional annotations were assigned via BLASTp (E-value ≤ 1e − 5) against Swiss-Prot[71], with motifs and domains identified by InterProScan (v.5.39)[72]. Gene Ontology (GO) terms and Kyoto Encyclopedia of Genes and Genomes (KEGG) pathway annotations were assigned accordingly. Non-coding RNAs were characterized as described in the Supplementary Note 2.

### Comparative genomics analysis
Synteny and whole-genome duplication events were evaluated using WGDI (v0.6.1)[33] for collinearity and $K_s$ distributions. Substitution rate variation was accounted for by employing four species trios: (1) *B. insignis, Adiantum capillus-veneris, Lygodium japonicum*; (2) *B. insignis, Ceratopteris richardii, L. japonicum*; (3) *B. insignis, Alsophila spinulosa, L. japonicum*; and (4) *B. insignis, Marsilea vestita, L. japonicum*. We standardized the synonymous substitution rates for each trio from divergence events between species to the focal species (*B. insignis*)[73].

For analyses of LTR retrotransposons (LTR-RTs), we obtained genomic data from seven available fern species—*Salvinia cucullata*[74], *Azolla filiculoides*[74], *Marsilea vestita*[75], *Adiantum capillus-veneris*[29], *Alsophila spinulosa*[28], *Adiantum nelumbodies*[76], and *Ceratopteris richardii*[27]—from publicly available datasets. LTR-RTs were identified uniformly in eight fern species using LTR_Finder[77], LTRharvest[78], and LTR_retriever[79]. We applied a phylogenetically informed calibration to account for lineage-specific rate heterogeneity[80]. For LTR retrotransposons, we estimated the corrected nucleotide substitutions ($K$) between paired 5' and 3' LTRs using LTR_retriever, which employs the Jukes-Cantor (JC69) model. The resulting $K$ values were converted into LTR insertion times using lineage-specific substitution rates calibrated on a phylogenetic framework, enabling cross-species comparisons. The results were visualized in a density plot using ggplot2 (v.3.5.1)[81].

To investigate evolutionary rates, we performed phylogenomic analyses on 16 representative fern species using 8,720 low-copy and 31 single-copy orthologs identified by Orthofinder (v.2.5.5)[82]. Multiple sequence alignments were performed in MAFFT (v.7.520)[83], trimmed by trimAL (v.1.4.rev15)[84]. A maximum likelihood (ML) phylogeny was then inferred with IQ-TREE (v.2.2.6)[85], employing 1,000 bootstrap replicates. Substitution rates ($d_S$, $d_N$) were estimated with PAML (v.4.8)[86], and a likelihood ratio test compared free-ratio and one-ratio models. The significance of differences in synonymous (dS) and nonsynonymous (dN) substitution rates among the predefined groups (single-copy and low-copy) was evaluated using analysis of molecular variance (AMOVA). Finally, relative rate tests (RRT) were performed in MEGA[87] on a concatenated protein alignment of 31 single-copy genes.

## Gene family identification and evolution

For gene family evolution, a phylogenetic tree was constructed across 16 taxa, including a bryophyte (*Physcomitrium patens*), a lycophyte (*Selaginella moellendorffii*), four seed plants, and ten ferns (including *B. insignis*). We rooted the resulting tree with *P. patens*, converted it into an ultrametric chronogram using r8s[88], and calibrated it against TimeTree (v.5.0)[89]. CAFE (v.4.2.1)[90] was used to identify significantly expanded or contracted gene families ($P < 0.05$). We characterized 11 gene families associated with the monolignol biosynthesis based on homology to *Alsophila spinulosa* genes. One-to-one orthologs in *B. insignis*, *A. spinulosa*, and *Sphaeropteris lepifera* were identified by reciprocal BLAST, and $K_a/K_s$ ratios were calculated in KaKs_Calculator (v.3.0)[91] to assess selective pressures on lignin-related genes.

## Resequencing, SNP calling, and filtering

Fresh leaves from 94 *B. insignis* individuals representing 29 geographic locations (Supplementary Data 1) were used for DNA extraction using the CTAB method[52]. Paired-end libraries were sequenced on the DNBSEQ-T7 platform, and raw reads were quality-filtered with fastp (v.0.22.0)[92]. Clean reads were aligned to the *B. insignis* reference genome using BWA-MEM[59]. The resulting SAM files were converted to BAM format and sorted with SAMtools[93]. PCR duplicates were removed via Picard (https://broadinstitute.github.io/picard/). Variants were called following the GATK (v.4.5.0.0) pipeline[94]. Low-confidence variants were removed using the following strict hard-filtering thresholds: QD < 2.0 || MQ < 40.0 || FS > 60.0 || SOR > 3.0 || MQRankSum < -12.5 || ReadPosRankSum < -8.0. Subsequently, a multi-step soft-filtering process was applied to generate multiple high-quality SNP datasets for downstream analyses (Supplementary Fig. 15). We define the high-confidence variants (core variants) as biallelic SNPs retained after joint genotyping and stringent hard/soft filtering. All downstream variant panels (Dataset 1-3) were derived from this core variant set (Supplementary Fig. 15).

## Population structure and genetic diversity analyses

To assess population structure, we generated 342,582 unlinked SNPs (minor allele frequency > 5%), which were analyzed with ADMIXTURE (v.1.3.0)[95] for K = 1 to K = 8 for cross-validation. Principal components analysis (PCA) was performed in PLINK (v.1.9)[96], and a neighbor-joining (NJ) phylogenetic tree was constructed using PHYLIP. A haplotype network was inferred with PopART[97] using the 95% statistical parsimony (TCS) network method (Supplementary Note 3). Nucleotide diversity (π), genetic differentiation coefficient ($F_{ST}$) and Tajima's *D* were estimated in 100-kb windows using pixy (v.1.2.10)[98] and VCFTools (v.0.1.16)[99]. Linkage disequilibrium (LD) decay was calculated via PopLDdecay (v.3.42)[100].

## Demographic history inference

For demographic history, we selected six samples ($\geq 20\times$ coverage) per major lineage to build diploid consensus sequences using BCFtools (v.1.14)[101]. PSMC[102] was run with default parameters (25 iterations, -N25, -r5, -p $4 + 25*2 + 4 + 6$) under a mutation rate of $1.25 \times 10^{-8}$ per site per generation and a 5-year generation time. In addition, SMC++ (v.1.15.2)[103] was performed to infer a finer-scale reconstruction of recent demographic histories within each lineage. To model complex demographic scenarios, we employed a stepwise, hierarchical approach using fastsimcoal2[104]. We initially modeled simpler histories with fewer parameters (e.g., divergence without migration), and progressively incorporated additional features such as continuous migration and pulse admixture. A total of 10 demographic and 5 differentiation models (Supplementary Fig. 21 and 22) were evaluated, each with 100 independent runs. The best-fitting scenario was selected based on the lowest AIC and ΔLhood (Supplementary Table 8–10). This stepwise strategy balances model realism with statistical robustness and mitigates potential identifiability issues arising from simultaneous estimation of many parameters.

To investigate gene flow, we generated a maximum likelihood drift tree using TreeMix (v.1.13)[105] with migration edges from 1 to 5, selecting the optimal number with OptM[106]. Finally, identity-by-descent (IBD) segments were identified using the BEAGLE algorithm[107].

## Estimation of inbreeding and genetic load

Individual heterozygosity was estimated with ANGSD (v. 0.937)[108] under a folded site frequency spectrum (fSFS) model. Runs of homozygosity (ROHs) were identified with PLINK (v.1.9)[96] using a minimum of 50 consecutive SNPs per 100-kb window. The inbreeding coefficient ($F_{ROH}$) was calculated as the proportion of the genome covered by ROHs[109,110].

Derived alleles were defined as those where more than 50% of individuals carried the same homozygous genotype[111]. SnpEff (v.4.3t)[112] was used to annotate missense and loss-of-function (LOF) variants. Missense mutations were further assessed using SIFT-4G[113] to classify potentially deleterious (DEL) substitutions, and the Grantham Score (GS $\geq 150$)[114] was applied to confirm deleterious missense variants. We calculated the proportion of homozygous DEL and LOF sites relative to the total number of homozygous plus heterozygous sites of each individual. Correlations among heterozygosity, inbreeding, and deleterious load (DEL and LOF) were visualized with SRplot[115]. We compared homozygous LOF variant frequencies inside and outside ROHs to evaluate the impact of inbreeding on genetic load. To quantify nonsynonymous and synonymous diversity, we identified 0- and 4-fold degenerate sites from the reference CDS using the degenotate pipeline (https://github.com/harvardinformatics/degenotate), extracted the corresponding population polymorphism data from variant call format (VCF) file, and calculated nucleotide diversity (π) at these sites with ANGSD[108].

## Selective sweep analysis

Selective sweeps were examined for two major lineages (YN and SC). We used a combined approach incorporating high $F_{ST}$ signals (top 5% in 500 kb windows, 5 kb step) and within-lineage sweeps identified by RaiSD (v.2.9)[116]. Candidate selective sweeps were defined where both methods overlapped, with the 99.99% quantile of μ serving as the threshold. Manhattan plots were generated in R using the qqman package[117].

## Genotype-environment association analyses

We retrieved 19 bioclimatic layers (1970–2000) from WorldClim (https://worldclim.org/) using ArcGIS v.10.8 for spatial extraction and retained seven uncorrelated variables ($|r| < 0.7$) via Pearson correlation: bio1, bio2, bio3, bio7, bio12, bio14, and bio15 (Supplementary Fig. 34). To identify a robust set of climate-adapted loci, we applied Latent Factor Mixed Models (LFMM) and Redundancy Analysis (RDA) to test genotype–environment associations (GEA). LFMM were conducted using three latent factors in the LEA package (v.3.16.0)[118] to account for population structure. Variants significantly associated (FDR < 0.05) with at least three environmental variables were considered outliers. RDA was performed using vegan (v.2.6-8)[119]. Significant variants were identified based on extreme loadings (±3.5 SD, equivalent to a two-tailed *P* value of 0.0005) along one or more RDA axes.

To disentangle the effects of isolation by distance (IBD) from isolation by environment (IBE), we conducted Mantel and partial Mantel tests separately on adaptive and neutral SNP datasets. Pairwise matrices of $F_{ST}/(1 - F_{ST})$ were tested for correlation with geographic and environmental distances using the vegan package.

## Genetic offset under future climate scenarios

Genetic offset (also termed genomic vulnerability or risk of non-adaptedness, RONA[49,50]) was estimated specifically using the adaptive SNPs. We applied generalized dissimilarity modeling (GDM)[120,121] with future climate projections (2081–2100) from WorldClim2.1 under two climate models (BCC-CSM2-MR and

GISS-E2-1-G) and two Shared Socioeconomic Pathways (SSP245 and SSP585) at 2.5 arcminutes resolution. Because the two climate models showed a strong correlation (Supplementary Fig. 35), BCC-CSM2-MR was used. Forward and reverse genetic offsets[120,122] were calculated to account for the potential effects of migration, then normalized across each channel (RGB) for spatial visualization. The local, forward, and reverse offsets were then mapped to the red, green, and blue channels, respectively. For each grid, we calculated genetic offsets and the maximum allowable migration distance and used Matplotlib (https://matplotlib.org/) to create a polar plot representing migration distance and direction.

### Reporting summary
Further information on research design is available in the Nature Portfolio Reporting Summary linked to this article.

### Data availability
Data supporting the findings of this work are available within the paper and its Supplementary Information files. The plant materials generated during the current study are available from the corresponding author upon request. All raw sequence datasets and genome assembly have been deposited at NCBI under the BioProject PRJNA1203580. The genome annotation files are available at Figshare (https://doi.org/10.6084/m9.figshare.28229507). Source data are provided with this paper.

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

## Acknowledgements

We thank Chao Feng (South China Botanical Garden, Chinese Academy of Sciences) for his assistance with sample sequencing and coordination, Weiping Zhang and Huiqin Yi (South China Botanical Garden, Chinese Academy of Sciences) for providing valuable discussions on recent advancements in population genomics, Zhengyu Zuo (Kunming Institute of Botany, Chinese Academy of Sciences), Yigang Song (Shanghai Chenshan Botanical Garden) and Qifei Yi (South China Botanical Garden, Chinese Academy of Sciences) for providing several molecular samples, Juan Li and Jiangping Shu. (National Orchid Conservation Center of China and the Orchid Conservation & Research Center of Shenzhen) for their help with gametophyte transcriptome sampling, and Guoen Ding and Zuoying Wei (South China Botanical Garden, Chinese Academy of Sciences) for accompanying the field sampling trip. This study was supported by the National Key R&D Program of China (No. 2024YFF1306600), Guangdong Flagship Project of Basic and Applied Basic Research (No. 2023B0303050001) and Basic research project independently deployed by South China Botanical Garden, Chinese Academy of Sciences (No. JCYJXM-202504).

## Author contributions

F.-G.W., M.K., Y.V.d.P. and Z.-Q.X. planned and coordinated the study. M.K. and Z.-Q.X. defined the major scientific objectives. Z.-Q.X.

performed the genome assembly, annotation, comparative genomics, and population genomic analyses. F.-G.W., M.K., Y.V.d.P. and Z.-Q.X. wrote the manuscript. L.D. and Y.-H.F. assisted with data analysis and provided suggestions on manuscript drafting. Y.J. and Z.-X.L. contributed to data visualization. Y.-H.Y., H.-F.C. and H.S. offered suggestions on the manuscript. A.-H.W., G.-H.Z. and Z.-Y.L. contributed to the sample collection and data analyses.

## Competing interests

The authors declare no competing interests.
