## [Peer Review file · Nature Communications]

Decoding the *Brainea insignis* genome informs fern evolution and conservation

Corresponding Author: Professor Faguo Wang

Version 0:

Reviewer comments:

Reviewer #1

(Remarks to the Author)

Xia et al presented the first fern genome from the eupolypod II clade, and characterized the history of whole genome duplication, repeat expansion, gene family evolution, and population demography. I found the results on WGD and repeats very interesting, which have broad implications in understanding the gigantic genomes in ferns. On the other hand, I'm somewhat skeptical about authors' take on the evolution of lignin genes. Without knowing exactly what kinds of unique lignin *Brainea* has, I feel the authors are overinterpreting their CAFÉ results. Furthermore, with their sampling, the expansion/contraction is not specific to *Brainea* per se, but the >100-million-years-old lineage leading to the species. I'm not an expert of population genomic analyses and cannot comment on that part of the manuscript.

Line-by-line comments below:

Line 123 and 147. The BUSCO scores are low, but I think it's because the authors used the embryophyta set, which is heavily based on angiosperm genomes. I'd suggest using the viridiplantae set, one that has more representations from non-seed plants.

Line 181-186. I don't think the data presented support that *Brainea* has had low substitution rate. From extended fig. 3, I can only see lower rate from Ks on single copy genes, and the rest look all the same. But in the text, it was stated that "*B. insignis* and its close relatives (*Woodwardia prolifera* and *Pronephrium simplex*) showed significantly lower dS and dN values than other ferns (Extended Data Fig. 3)." I just don't see the pattern and what stats were done to support this?

Line 207-223. I think the interpretation here is problematic. The inferred expansions/contractions did not necessarily happen to *B. insignis* specifically, but the lineage leading to it. The closest species on this CAFÉ analysis was >100 million years apart. Therefore, there is no evidence that the expanded lignin biosynthesis genes "plays a crucial role in the persistence of this monotypic genus"

Line 330. "To identify genes underlying lineage-specific adaptations" – this to me sounds like you have evidence for local adaptations. But I'm not sure if that evidence is presented.

Line 385-386. I'm not sure I follow how slow substitution reflects "ecological adaptability".

Fig. 1b. There is a sudden spike in gene density at Chr04? Curious to know what that is about.

Fig. 3. It would be nice to know the distribution range of this species, so that one can assess if the sampling is representative and capturing the geography.

Reviewer #2

(Remarks to the Author)

To the Authors and Editors,

Xia et al. provide a well-written manuscript of a well-executed study detailing the assembly and annotation of the *Brainea*

insignis genome, as well as a population genomic assessment of the genetic diversity and demographic history of this endangered fern species. After several readings of the manuscript and supplemental information, I cannot find any major flaws in their study design, methods, or data analyses related to the genome sequencing, assembly, annotation, or populations genomic analyses. The work (with one exception) meets the standards of the field, and I feel that the methods are of sufficient detail to be reproducible. I will refrain from assessing their evaluation of genetic offset under future climate scenarios—this is too far afield from my expertise. I do, however, take some issue with some of their wording and interpretation of results, and I will detail a few examples at the end of this review.

This work has substantial significance for the fields of comparative plant genomics, fern evolution and population genetics, and conservation genomics. In my opinion, the greatest impact of this publication is that it provides the first chromosome - level assembly for the Eupolypod II ferns, representing approximately 20% of extant fern diversity. It fills a major gap in genomic resources for ferns. However, *Brainea insignis*, a monotypic genus and “living fossil” akin to *Ginkgo biloba* and *Cercidiphyllum japonicum*, is an odd, atypical choice to represent this group. Perhaps, this choice reflects the authors’ focus on conservation, rather than comparative genomics for this study?

Nonetheless, this genome assembly is a novel and needed resource for meaningful comparative genomics investigations of chromosome number, genome size, genic content, and rates of evolution relative to other ferns and to other major plant lineages, especially their sister group the seed plants. The authors briefly touch on this importance throughout the manuscript by confirming an ancestral WGD for the core leptosporangiate ferns, comparing intron sizes/exon numbers across ferns, illustrating the positive relationship between TE content and genome size in ferns, assessing rates of evolution among ferns, etc. In my opinion, the appeal and application of this manuscript would be broadened by refocusing and expanding discussion of these topics.

A second major contribution of this work is a thorough assessment of population genetic structure in a biogeographic context, demographic history, gene flow, and signatures of local selection. The study of population genomics in ferns, particularly employing whole genome resequencing, is essentially non-existent and has been long hampered by a lack of genomic resources. I can think of only one other study (Yi et al. 2024. *The Plant Journal* 120(4):1522-35) that utilized a comparable genomic-scale dataset to estimate signatures of inbreeding and mutational load in a fern. I believe this manuscript will become a crucial resource/protocol for other researchers interested in population-level inquiries in ferns, in particular, for address many long-standing but previously intractable questions related to breeding systems in organisms that are spore dispersed and have independent sporophyte/gametophyte life phases.

A third contribution of this work is that it provides further support for the independent evolution of lignin syringly “S” units, mainly reported from angiosperms, in *B. insignis*. This finding builds on a recent *Nature Plants* publication (Ali et al. 2024. <https://doi.org/10.1038/s41477-025-684 01978-y>), indicating that lignin S-units have evolved repeated across land plants, and that ferns have independently evolved mechanisms of structural support and vasculature in terrestrial environments.

As mentioned in the first paragraph, there are several instances where I feel the authors inaccurately depicted background information or have interpreted their results too liberally. Below are three examples, that could be amended:

1-Line 60 “Due to their phylogenetic position bridging non-vascular bryophytes and seed-bearing vascular plants...” This may seem overly picky, but I think it misrepresents the placement/informativeness of ferns in plant evolution. Ferns do not “bridge” or have some type of “intermediate” phylogenetic placement. (As the authors know) ferns are the sister group to seed plants, and share a common ancestor with seed plants, not shared with bryophytes. Please clarify for accuracy.

2-Line 130 “These results collectively underscore the high contiguity, consistency, and completeness of our *B. insignis* assembly.” I am concerned about the authors’ conclusion of completeness given the BUSCO score of 81.5% complete recovery (77.7% based on annotation assessment, Supplementary Table 3). Despite having high LAI and CRAN metrics, this is a very low BUSCO score (The expected standard for the field is usually 95% complete or higher). At minimum, the authors need to address the possible reasons for the low BUSCO recovery and discuss whether it is an artifact of their assembly or possibly reflecting a biological reality (unlikely).

3-Line 187 “relative rate tests suggest that *B. insignis* has the slowest evolutionary rate among core leptosporangiate ferns (Supplementary Table 7).” Given the information in Supplementary Table 7, this is an overexaggeration. It appears that *B. insignis* does not have a significantly slower rate of evolution (in 31 single copy genes) relative to several other core leptosporangiate ferns, including *L. chinensis* and *D. repens*. Please amend this statement and temper your conclusion that “this slow evolutionary rate may have bolstered genomic stability, enabling ...”.

Thank you for the opportunity to review this manuscript.

Version 1:

Reviewer comments:

Reviewer #1

(Remarks to the Author)

I appreciate the authors’ response to my previous comments. I’m however still skeptical about their take on the evolution of

lignin genes, as well as their analyses on evolutionary rates. Please see my comments below. I'm not an expert in population genomic analyses and cannot comment on that part of the manuscript.

Line 184-186. First, I won't consider *Pronephrium simplex* a "close relative". It's in a different family. Second, I still don't understand why only single copy genes show rate difference, and not the low copy genes. Importantly, since single copy genes only account for a small proportion of the genome, can they really represent the evolutionary rate of whole genomes?

Line 189-192. The relative rate tests similarly have issues. I am not sure to what extent 31 single copy genes are representative of this species. Further, with *Psilotum nudum* as the outgroup, the timescale here is vast. In other words, is the rate difference truly specific to *Brainea* or the hundreds of million years of evolution leading to it? I am not certain the analyses here are valid.

Line 200-201. "larger fern genomes tend to harbor a higher proportion of LTR-RTs with earlier insertion time". How can you accurately estimate and compare insertion times when there is rate heterogeneity among fern lineages (a point you are trying to make above)?

Line 213-216. The vast majority of the genes are evolving under purifying selection ($w < 1$) and I do not believe this can be used to support "strong functional constraint on these lignin-related genes across these fern lineages." In addition, *Brainea* and *Sphaeropteris* span deep evolutionary time, during which arborescence only accounts for a small blip in time – does this calculation actually mean anything?

Line 219-222. Another possibility is that the expansion of these gene families has nothing to do with woodiness!

Line 225-228. What's the evidence for "convergent evolution of lignin specialization"? Do we know what types of lignin *Brainea* actually have? There are a lot of vague associations and assumptions in this section.

Minor comments:

Line 60. "ferns share a common ancestor with them" – Every organism on this planet shares a common ancestor with another, so this statement is not needed.

Line 89. Who is "them" in "the broader biodiversity that depends on them"?

Line 127. "Clipping-based Revealing Assembly Quality" should be "Clipping Reveals Assembly Quality" or "Clipping information for Revealing Assembly Quality"?

Reviewer #2

(Remarks to the Author)
Dear Editors and Authors,

It is my opinion that Li et al. have satisfactorily addressed both reviewers' comments in a manner that improves the execution of the study and better highlights its immediate and broader impacts. The methodologies are sound, of the standards of the field, reproducible, and noteworthy. As described in my earlier review, I believe this manuscript is a well-written and valuable contribution as the first complete Eupolypod II fern genome assembly and as a rare example of using whole genome sequencing to investigate population level diversity for a fern species. I especially appreciate that the authors have broadened their discussion of the importance of this genome for investigating fern genome evolution. Of particular note, is their result that large genome size in the polypods seems largely dictated by TE expansion, rather than by WGD followed by slow rates of diploidization – this is important because it counters the existing paradigm for how ferns have accrued large chromosome numbers and genome sizes relative to angiosperms. I believe this study will also serve as a useful and well-cited model for population genomics approaches in ferns, a group historically underrepresented in population genomics because of limited genomic resources and large population size.

One caveat is that I wish the manuscript had been assigned to a reviewer with expertise in the evaluation of genetic offset under future climate scenarios. I feel that this aspect of the manuscript largely went unevaluated and might benefit from an expert's opinion.

Reviewer #3

(Remarks to the Author)

Xia et al. present a study that assembled a de novo genome of a cycad fern and conducted population genomic analyses based on 94 individuals. Although the dataset is substantial, primarily due to the species' large genome size, I find the novelty of the analysis somewhat underwhelming. Below, I outline several concerns and suggestions that could enhance the study's relevance and originality, particularly in comparison to other population genomic studies on species with smaller genome sizes.

Main Comments

1. Importance of the Genome and Comparative Analysis: While the de novo genome assembly of this endangered species

within the Eupolypods II clade is significant, the genomic analyses presented in the study feel overly general. Beyond the basic genomic structure in Figure 1, Figure 2 highlights WGD (whole-genome duplication) events and phylogenetic relationships across diverse species. It would be more impactful to compare this genome with other published fern genomes to examine genome collinearity and divergence among closely related species. A more detailed analysis could investigate unique transposable elements (TEs) or gene expansions/contractions specific to this species, as well as large-scale structural variations in the genome. Linking these variations to associated genes and their functional enrichment would provide deeper insight into this species' unique genomic features.

2. Integration of Demographic, Divergence, and Gene Flow Models: The study constructs demographic, differentiation, and gene flow models separately. However, these processes are interrelated and should ideally be integrated into a cohesive framework. Typically, divergence models incorporate demographic and gene flow histories to provide more accurate and comprehensive inferences. Analyzing these processes separately can lead to confounding effects and compromise the accuracy of the results. I suggest the authors revisit their fastsimcoal analysis, integrating demographic and gene flow scenarios into unified models for better reliability and interpretability.

3. Genetic Load and Population Decline: One of the study's key contributions is its emphasis on the endangered status of this species and the impact of population decline on genetic variation and genetic load. However, the current analysis only compares genetic load across the three genetic groups, which exhibit minimal differences and provide limited insight into the extent of genetic load accumulation. To enhance this analysis, the authors could employ established methods for measuring genetic load, such as the 0-fold/4-fold ratio or the homozygous/heterozygous counts of loss-of-function (LoF) and deleterious mutations. Furthermore, comparing the genetic load of this species to data from other studies on species with similarly small population sizes or those in an endangered state would offer a more meaningful context. This comparison could help clarify the severity of the genetic load accumulation in this fern species and its relative position on the spectrum of endangerment.

4. Issues with the Ratio of Nonsynonymous to Synonymous Diversity: On line 312, the authors report a nonsynonymous-to-synonymous diversity ratio close to 1, which suggests a potential calculation error. I recommend carefully reviewing the methodology used to compute this ratio. If the result is accurate, it represents an unusual or unique finding, as such a high value (π_0 -fold/ π_4 -fold) is rarely observed. In that case, further investigation is necessary to determine the underlying causes of this pattern.

5. Genetic Vulnerability to Climate Change: The study includes an analysis of genetic vulnerability to future climate change, but this is now a fairly standard approach. It's important to critically assess whether such methods are suitable for this species and whether they provide accurate predictions. The genetic offset analysis assumes environmental adaptation, but in this case, geographic distance is strongly correlated with environmental distance. This makes it difficult to disentangle the effects of selection-driven environmental adaptation from neutral genetic drift due to geographic separation. To address this issue, the authors could focus on specific identified adaptive variants and test whether they are associated with environmental variables, geographic differences, or a combination of both. Differentiating these factors is essential for understanding the role of environmental adaptation before conducting genetic offset analyses.

6. Quantification of Genetic Offset Results: The genetic offset analysis currently presents results in the form of a colored plot, which makes it difficult to pinpoint which populations are more vulnerable. I recommend quantifying the offset values (e.g., local, forward, and reverse offsets) for each population or genetic cluster and visualizing them using boxplots or similar approaches. This would allow readers to clearly identify which populations are better adapted and which are more vulnerable to future climate change.

Minor Comments

1. On line 241, the term core variants is unclear. How is it defined? Providing a clear definition would improve clarity.
2. In general, the Results section contains several ambiguous statements, partly because the methods are described later. Restructuring or clarifying these sections would make the study easier to follow.

Version 2:

Reviewer comments:

Reviewer #1

(Remarks to the Author)

I believe the authors have nicely addressed most of my previous concerns. I found the new synteny comparison with *Alsophila* very interesting. I do want to point out that computing "Ks" between LTR repeats is not valid. Ks is synonymous substitution and would only apply to protein-coding genes. How can you calculate Ks for repeats?

Reviewer #3

(Remarks to the Author)

I appreciate the efforts of the authors in addressing my previous comments, most of which have been resolved. Thank you for your efforts. However, I have two minor comments below:

- (1) Although the analysis of genetic load in ferns is limited, it would still be interesting to compare the genetic load levels with

those of other plant species, such as angiosperms. I believe genetic load data are available for more than 20 species, and it would be valuable to compare the genetic load of endangered ferns with other endangered plant species across phylogenetic positions.

(2)The forward and reverse differences across different populations shown in Supplementary Figure 25 are difficult to distinguish, likely because the local offset values are too high. I suggest that instead of using the same value range for all the plots, each figure should use its own value range for plotting the offset. This adjustment would help to clearly visualize the landscape distribution of offset values and make it easier to differentiate between forward and reverse offsets.

Version 3:

Reviewer comments:

Reviewer #3

(Remarks to the Author)

I appreciate the authors for addressing my concerns. I have no further comments and look forward to seeing this work published soon.

Response to Reviewer's comments

**Reviewer #1 (Remarks to the Author):**

Xia et al presented the first fern genome from the eupolypod II clade, and characterized
the history of whole genome duplication, repeat expansion, gene family evolution, and
population demography. I found the results on WGD and repeats very interesting, which
have broad implications in understanding the gigantic genomes in ferns. On the other
hand, I'm somewhat skeptical about authors' take on the evolution of lignin genes.
Without knowing exactly what kinds of unique lignin *Brainea* has, I feel the authors
are overinterpreting their CAFÉ results. Furthermore, with their sampling, the
expansion/contraction is not specific to *Brainea* per se, but the >100-million-years-old
lineage leading to the species. I'm not an expert in population genomic analyses and
cannot comment on that part of the manuscript.

**Response:** Thank you very much for your insightful and thoughtful feedback on our
manuscript. We are glad to hear that you found our results on whole genome duplication
(WGD) and repeat expansion interesting. Regarding the evolution of lignin genes, we
understand your skepticism. We acknowledge that the exact composition of lignin in
*Brainea insignis* remains unknown and agree that this limits definitive interpretations.
We also recognize that the gene family expansion/contraction patterns observed in our
CAFÉ analysis may reflect deep evolutionary events along the >100-million-year-old
lineage leading to *B. insignis*, rather than changes specific to the species itself.

To address this concern, we conducted a revised CAFÉ analysis with denser taxon
sampling focused on lineages more closely related to *B. insignis* (see Response Fig. 1a).
This updated analysis better resolves recent divergence events and helps distinguish
lineage-specific shifts from more ancient trends. Notably, we still detected 353
significantly expanded gene families ($p < 0.05$) in *B. insignis* that are enriched in
biological processes related to “cell wall organization” and “lignin metabolic process”
(Response Fig. 1b). In contrast, a parallel analysis in the herbaceous fern *Adiantum*
*capillus-veneris*, which lacks lignified woody structures, revealed no enrichment in
lignin-related pathways among its significantly expanded gene families (Response Fig.
1c). These results strengthen our confidence that lignin biosynthesis-related gene
families have indeed undergone expansion in *B. insignis*, potentially reflecting the
species' adaptation to a woody, tree-fern growth habit. We have revised the manuscript
to clarify that these expansions are inferred for the lineage leading to *B. insignis* and to
avoid overstating their importance without direct chemical evidence of lignin
composition in this species (Manuscript Lines 208-211).

Response Fig. 1 | CAFE and GO enrichment analysis

Line 123 and 147. The BUSCO scores are low, but I think it's because the authors used
 the embryophyta set, which is heavily based on angiosperm genomes. I'd suggest using
 the viridiplantae set, one that has more representations from non-seed plants.

**Response:** We sincerely appreciate this insightful suggestion regarding our BUSCO
 analysis. As correctly noted, our initial use of the embryophyta_odb10 dataset, which
 is biased toward angiosperms and may not be ideal for fern genome. In response, we
 have re-evaluated both our genome assembly and annotation using the more inclusive
 viridiplantae_odb12 dataset (updated July 1, 2025), which include a broader
 representation of non-seed plants. The updated BUSCO results (presented in Response
 Table 1 and incorporated into the manuscript) show a substantial improvement in
 completeness. For example, the genome assembly's BUSCO completeness increased
 from ~81.5% to 97.4% using the viridiplantae dataset, indicating that the earlier lower
 score was indeed an artifact of the reference dataset choice rather than genome quality.
 We thank the reviewer for this valuable suggestion, which has significantly improved
 the rigor of our genomic assessment.

**Response Table 1 | Updated BUSCO completeness assessment using the**
 **viridiplantae odb12 dataset**

Evaluation Type	Complete (C)	Single-copy (S)	Duplicated (D)	Fragmented (F)	Missing (M)	Total BUSCOs (n)
Genome Assembly	97.4%	89.9%	7.5%	2.1%	0.5%	822
Gene Annotation	84.2%	79.1%	5.1%	12.7%	3.2%	822

Line 181-186. I don't think the data presented support that *Brainea* has had low
 substitution rate. From extended fig. 3, I can only see lower rate from Ks on single copy
 genes, and the rest look all the same. But in the text, it was stated that "*B. insignis* and
 its close relatives (*Woodwardia prolifera* and *Pronephrum simplex*) showed
 significantly lower dS and dN values than other ferns (Extended Data Fig. 3)." I just
 don't see the pattern and what stats were done to support this?

**Response:** We appreciate the reviewer's careful evaluation of our substitution rate
 analysis. We agree that our original wording was too strong. The statement that "*B.*
 *insignis* has the lowest substitution rate among core leptosporangiate ferns" was an
 overgeneralization and did not adequately convey the nuances of the data. In reality,
 different genes evolve under different selective pressures, so it is unlikely that *B.*
 *insignis* would exhibit uniformly lower substitution rates across all genes compared to
 other ferns. Furthermore, our current sampling of fern species remains limited and may
 not fully capture the range of substitution rate variation in this clade.

Nevertheless, several lines of evidence suggest that *B. insignis* generally evolve
 somewhat slower than many other core leptosporangiate fern: 1) WGD peak
 comparison: When analyzing the synonymous substitution (Ks) peaks corresponding to
 the shared ancient WGD event in ferns, *B. insignis* shows a significantly lower peak
 value (~1.7) compared to the fern *Adiantum capillus-veneris* (~2.25). This indicates
 slower accumulation of substitutions in *B. insignis* since that duplication event. 2)
 Relative substitution rate analyses: Our relative substitution rate tests consistently
 indicate slower molecular evolution in *B. insignis* in many comparisons. However, the
 differences are not statistically significant when *B. insignis* is compared to certain
 species with similarly slow rate (e.g., *A. spinulosa*, *P. simplex*, and *B. hekouensis*). We
 have now performed additional statistical tests and included them in Extended Data Fig.
 3 to quantify these differences and their significance. 3) Single-copy gene Ks values:
 As the reviewer noted, *B. insignis*'s single-copy genes exhibit lower Ks values relative
 to most other ferns examined. We consider this particularly informative because
 synonymous substitution rates (dS) are less subjected to selection and other

confounding factors, providing more direct indicator of the neutral mutation rate. The
consistently lower dS for *B. insignis* single-copy genes suggests a broadly slower base
substitution rate.

We have revised the manuscript text to temper our language and avoid absolute claims
(Manuscript Lines 186-189). Additionally, we added a more comprehensive statistical
comparison of dN and dS across species in Extended Data Fig. 3. This update provides
a clearer visualization and statistical support to back the statement that *B. insignis* tends
to have a slower evolutionary rate than many core leptosporangiate ferns (while
acknowledging it is not the slowest in every comparison).

Line 207-223. I think the interpretation here is problematic. The inferred
expansions/contractions did not necessarily happen to *B. insignis* specifically, but the
lineage leading to it. The closest species on this CAFÉ analysis was >100 million years
apart. Therefore, there is no evidence that the expanded lignin biosynthesis genes “plays
a crucial role in the persistence of this monotypic genus”

**Response:** Thank you for this important clarification. We agree that the expansion and
contraction events identified by CAFÉ likely occurred in an ancestral lineage (over 100
Mya) leading to *B. insignis*, rather than in the modern species itself. Consequently, our
original wording in the manuscript attributing these changes specifically to *B. insignis*
(and suggesting they “play a crucial role in the persistence of this monotypic genus”)
was potentially misleading. We have revised that statement in the manuscript to be more
precise and cautious, removing any implication that we have direct evidence for such a
cause-and-effect in *B. insignis*.

To more accurately trace these gene family dynamics, we performed a new CAFÉ
analysis with denser taxon sampling to shorten the phylogenetic branch lengths
between *B. insignis* and the comparison species. By including additional closer relatives,
we reduced the divergence times and thus minimized attributing the gene family
changes to deep ancestral nodes. The updated results (included in Response Fig. 1 as
referenced above) still indicate an expansion of lignin biosynthesis-related gene
families on the branch leading to *B. insignis*. This gives us more confidence that the
trend is relevant to *B. insignis*'s lineage. However, we now discuss these results with
appropriate context: the expansion likely occurred in an ancestor and may have been
retained in *B. insignis* due to its woody growth form. In the revised Manuscript, we
have tempered our language about the significance of this finding for *B. insignis*'s
persistence, noting it as a hypothesis rather than a confirmed mechanism.

Line 330. “To identify genes underlying lineage-specific adaptations” – this to me
sounds like you have evidence for local adaptations. But I’m not sure if that evidence
is presented.

**Response:** We appreciate the reviewer’s careful reading of our manuscript. You are
correct that our original phrasing was premature. Simply observing long-term isolation
and genetic structure among the YN, SC, and VN lineages (from our population
demographic analyses) does not, on its own, prove local adaptation. We agree that
evidence for local adaptation should come from explicit tests, such as detecting
selective sweeps or genotype–environment correlations, rather than from population
structure alone.

In response to this concern, we have removed and rephrased any speculative statements
about “lineage-specific adaptations” that were not directly supported by evidence. More
importantly, we strengthened our analysis to present comprehensive evidence
supporting local adaptation through: 1) Selective sweep analysis: We conducted
selective scans within each lineage (especially the SC and YN groups) and identified
sets of genes that show signatures of recent positive selection unique to each lineage.
Several of the lineage-specific candidate genes are enriched for functions that make
biological sense in the context of their different habitats. This suggests that each lineage
has experienced adaptive divergence; and 2) Genotype-environment association (GEA):
We performed GEA analyses and detected significant isolation-by-distance (Mantel’s r
= 0.4301, $p < 0.05$) and isolation-by-environment (Mantel’s $r = 0.4058$, $p < 0.05$)
patterns in core adaptive loci. Redundancy analysis further demonstrated three distinct
genetic clusters corresponding to each lineage (YN, SC, VN), with admixed
populations occupying intermediate positions. Notably, the YN lineage’s genetic
composition is strongly correlated with high diurnal temperature range (bio2), while
SC associated with low bio2, high annual mean temperature (bio1), and high
precipitation in the driest month (bio14). The VN lineage formed a unique
environmental cluster, with admixed populations showing heterogeneous
environmental signatures.

Together, these results provide robust evidence that *B. insignis* has undergone local
adaptation, with each lineage’s genome shaped by the specific environmental
conditions of its region alongside the effects of geographic isolation. We have clarified
the language to differentiate clearly between observed evidence of local adaptation (as
above) and any remaining speculative inference. We believe this addresses the concern
by replacing unsupported implication with concrete data.

Line 385-386. I'm not sure I follow how slow substitution reflects "ecological
adaptability".

**Response:** Thank you for this insightful comment. The connection we intended to draw
between "slow substitution rates" and "ecological adaptability" is rooted in the idea that
some long-lived or "living fossil" plant species occupy exceptionally stable niches. For
such species, a slow rate of molecular evolution is often a consequence of their enduring
stability and success in an unchanging environment, rather than a cause of adaptability
160 per se. In other words, if a species like *B. insignis* has persisted for millions of years in
a relatively stable ecological niche, it may not require rapid genetic change to remain
well-adapted. Thus, its substitution rate can appear low because it has already achieved
a highly adapted state and continues to thrive without drastic genomic innovation. For
example, Xiang et al. (2024) reported that the living fossil tree *Euptelea pleiosperma*
exhibits an unusually low substitution rate, which the authors associated that specie's
long-term occupation of a conserved niche. Therefore, we suggest that in *B. insignis*,
the observed slow substitution rate might be a signal that it is already well-adapted to
its ecological niches and does not need rapid genetic turnover to cope with its
environment. To avoid misunderstanding, we have rephrased this discussion in the
manuscript. We now clarify that a slow substitution rate is interpreted as a sign of long-
term persistence in stable niches (Manuscript Line 391), and we avoid using the term
"ecological adaptability" which was confusing.

**Reference:** Xiang, et al. Genomic data and ecological niche modeling reveal an
unusually slow rate of molecular evolution in the Cretaceous Eupteleaceae. *Science*
*China (Life Sciences)* 67.4(2024):803-816.

Fig. 1b. Is there a sudden spike in gene density at Chr04? Curious to know what that is
about.

**Response:** Thank you for your keen observation. Indeed, our genome assembly shows
a sudden spike in gene density on Chr04. We speculate that this localized increase in
gene number is due to gene duplication events (possibly tandem duplications) or other
adaptive processes that lead to a concentrated proliferation of related genes in this
region.

To investigate this feature further, we conducted a detailed analysis of the genomic
region underlying this spike. Our analysis localized the spike to a 1.2 Mb interval
(Chr04:106.2-107.4 Mb) encompassing 91 annotated genes. Strikingly, many of these
genes are closely related: our whole-genome collinearity analysis identified 79 gene
pairs within this interval that reside in local syntenic blocks (indicative of recent
duplications). Additionally, 32 of these genes in this cluster encode proteins with

multiple conserved domains (e.g., Pfam:Photo_RC, Pfam:Oxidored_nitro), with many
 sharing identical or highly similar functional domains (Response Fig. 2). These findings
 strongly suggest that this sudden spike region on Chr04 results from tandem gene
 duplication events that expanded a family of gene with related functions.

 **Response Fig. 2 | Conserved Protein Domains in the Chr04 Gene Cluster**

Fig. 3. It would be nice to know the distribution range of this species, so that one can
 assess if the sampling is representative and capturing the geography.

**Response:** Thank you for this valuable suggestion. *Brainea insignis* is primarily
 distributed across tropical Asia, with documented occurrences spanning from 98.5°E to
 126.4°E in longitude and from 7.8°S to 25.5°N in latitude, according to GBIF (Global
 Biodiversity Information Facility, July 24, 2025) records. This distribution
 encompasses regions across Southeast Asia, extending from continental Asia (including
 known occurrences in China’s Yunnan) through the Indonesian archipelago. In this
 study, we sampled 94 individuals from 32 wild populations covering a broad portion of
 *B. insignis*’s range. The samples span a broad longitudinal range (97.72°E-117.98°E)
 across Southeast Asia, with Vietnam’s representative population sampled at 12.19°N.

We acknowledged that our sampling does not cover the full geographic extremes of *B.*
 *insignis*’ distribution. However, we would like to highlight the significance of this effort
 within the field of fern genomics. Given that *B. insignis* possesses an exceptionally
 large genome (~8 Gb)—ranking among the largest fern genomes ever sequenced at a
 population scale—this study represents one of the most comprehensive whole-genome
 resequencing initiatives conducted in ferns to date. We are confident that our sampling

strategy captured the major geographic and environmental diversity of *B. insignis*
across its core range, and thus provides robust insights into the species' overall genetic
structure and demographic history.

In the revised manuscript, a comprehensive distribution map representing the full
known occurrences of *B. insignis* has been shown as Supplementary Fig. 9. As shown
below:

**Supplementary Fig. 9 | A total of 178 non-redundant occurrence records of *Brainea***
***insignis*.**

Red, blue, and black points represent sampling locations from this study, occurrence
records from NSII (National Specimen Information Infrastructure, China) and CVH
(Chinese Virtual Herbarium), and GBIF (Global Biodiversity Information Facility)
records, respectively.

**Reviewer #2 (Remarks to the Author):**

To the Authors and Editors,

Xia et al. provide a well-written manuscript of a well-executed study detailing the
assembly and annotation of the *Brainea insignis* genome, as well as a population
genomic assessment of the genetic diversity and demographic history of this
endangered fern species. After several readings of the manuscript and supplemental
information, I cannot find any major flaws in their study design, methods, or data
analyses related to the genome sequencing, assembly, annotation, or populations
genomic analyses. This work (with one exception) meets the standards of the field, and
I feel that the methods are sufficiently detailed to be reproducible. I will refrain from
assessing their evaluation of genetic offset under future climate scenarios – this is too
far afield from my expertise. I do, however, take some issue with some of their wording

and interpretation of results, and I will detail a few examples at the end of this review.
This work has substantial significance for the fields of comparative plant genomics,
fern evolution and population genetics, and conservation genomics. In my opinion, the
greatest impact of this publication is that it provides the first chromosome -level
assembly for the Eupolypod II ferns, representing approximately 20% of extant fern
diversity. It fills a major gap in genomic resources for ferns. However, *Brainea insignis*,
a monotypic genus and “living fossil” akin to *Ginkgo biloba* and *Cercidiphyllum*
*japonicum*, is an odd, atypical choice to represent this group. Perhaps, this choice
reflects the authors’ focus on conservation, rather than comparative genomics for this
study? Nonetheless, this genome assembly is a novel and needed resource for
meaningful comparative genomics investigations of chromosome number, genome size,
genic content, and rates of evolution relative to other ferns and to other major plant
lineages, especially their sister group the seed plants. The authors briefly touch on this
importance throughout the manuscript by confirming an ancestral WGD for the core
leptosporangiate ferns, comparing intron sizes/exon numbers across ferns, illustrating
the positive relationship between TE content and genome size in ferns, assessing rates
of evolution among ferns, etc. In my opinion, the appeal and application of this
manuscript would be broadened by refocusing and expanding discussion of these topics.

**Response:** We greatly appreciate the reviewer’s positive assessment of our study and
the insightful suggestions for improving the manuscript. We also acknowledge that *B.*
*insignis* is a monotypic genus and something of a “living fossil”, which indeed makes
it an unusual representative of Eupolypod II. We chose *B. insignis* partly because of its
conservation importance - it is endangered and thus of high conservation priority - and
partly because of its pivotal phylogenetic position as one of the most divergent core
leptosporangiate ferns. This dual rationale (conservation and comparative significance)
is now clarified in the Introduction of the revised manuscript (Manuscript Lines 86-89).

Crucially, while atypical, its genome provides a critical reference for understanding the
genomic stasis and divergence in ancient lineages. For example, our results support that
the core leptosporangiate ferns (which include Eupolypod I and II clades, together ~60%
of extant ferns) did not undergo any additional whole-genome duplications (WGDs)
beyond the ancient WGD shared by all core leptosporangiates. Instead, other factors –
such as the timing of lineage divergence and especially the accumulation of repetitive
elements – appear to have had a greater influence on genome size variation among these
ferns. We briefly touched on these points in the original manuscript, but we agree with
the reviewer that they warrant a more prominent discussion.

Given the current state of available genomic data, our ability to conduct fine-scale
comparative analyses was somewhat limited (the closest relative genomes available,

*Ceratopteris richardii* and *Adiantum capillus-veneris*, diverged from *B. insignis* >100
million years ago). We have proceeded with caution to avoid overinterpreting
differences that could simply be due to deep evolutionary divergence. Nonetheless, the
increasing availability of fern genomes will enable large-scale and in-depth
comparative genomic analyses in the future, and such broader comparisons—including
genome size evolution, TE dynamics, evolutionary rate variation, and synteny across
ferns—would strengthen our study.

A second major contribution of this work is a thorough assessment of population genetic
structure in a biogeographic context, demographic history, gene flow, and signatures of
local selection. The study of population genomics in ferns, particularly employing
whole genome resequencing, is essentially non-existent and has been long hampered
by a lack of genomic resources. I can think of only one other study (Yi et al. 2024. The
Plant Journal 120(4):1522-35) that utilized a comparable genomic-scale dataset to
estimate signatures of inbreeding and mutational load in a fern. I believe this manuscript
will become a crucial resource/protocol for other researchers interested in population-
level inquiries in ferns, in particular, for address many long-standing but previously
intractable questions related to breeding systems in organisms that are spore dispersed
and have independent sporophyte/gametophyte life phases.

**Response:** We greatly appreciate that you consider our population-genomic analysis
of *B. insignis* as a valuable resource for the community. As you pointed out, whole-
genome resequencing studies in ferns are extremely scarce, primarily due to the
historical lack of reference genomes and the computational challenges posed by the
typically large fern genomes. In fact, *B. insignis*'s genome (~8 Gb) is among the largest
fern genomes tackled with population resequencing to date, and this required us to
overcome substantial technical and computational hurdles (e.g. handling massive
amounts of sequencing data and repetitive sequences).

By demonstrating that whole-genome population studies are feasible in ferns, we hope
to inspire and enable more studies on fern population genetics—particularly to address
the unique questions you mentioned, like how breeding systems and life-cycle
characteristics of ferns influence their genetic diversity and inbreeding levels.

A third contribution of this work is that it provides further support for the independent
evolution of lignin syringyl “S” units, mainly reported from angiosperms, in *B. insignis*.
This finding builds on a recent Nature Plants publication (Ali et al.
2024. <https://doi.org/10.1038/s41477-025-68401978-y>), indicating that lignin S-units
have evolved repeatedly across land plants, and that ferns have independently evolved
mechanisms of structural support and vasculature in terrestrial environments.

**Response:** We appreciate the reviewer’s recognition of the aspect of our study. Indeed,
our results suggest that *B. insignis* (and possibly other ferns) have unique expansions
in certain lignin-biosynthesis gene families, which could relate to the production of
lignin S-units or analogous compounds for mechanical support. The recent finding by
Ali et al. (2025, Nature Plants) are highly relevant, as they showed that syringyl lignin
units—once thought to be exclusive to angiosperms—have evolved independently in
multiple plant lineages. This independent evolution of lignin chemistry and vascular
support strategies in ferns versus seed plants provides important context for our
observations. The evolution of lignin S-units in ferns remains an intriguing open
question. Our study not only supports the idea of independent lignin pathway evolution
but also sets the stage for future research to explore how ferns achieve their structural
support.

As mentioned in the first paragraph, there are several instances where I feel the authors
inaccurately depicted background information or have interpreted their results too
liberally. Below are three examples that could be amended:

Line 60 “Due to their phylogenetic position bridging non-vascular bryophytes and seed-
bearing vascular plants...” This may seem overly picky, but I think it misrepresents the
placement/informativeness of ferns in plant evolution. Ferns do not “bridge” or have
some type of “intermediate” phylogenetic placement. (As the authors know) ferns are
the sister group to seed plants, and share a common ancestor with seed plants, not shared
with bryophytes. Please clarify for accuracy.

**Response:** We thank the reviewer for pointing out this issue. You are absolutely correct;
our original phrasing was inaccurate. Ferns are not an intermediate between bryophytes
and seed plants; rather, ferns are the sister group to seed plants. This means ferns and
seed plants share a common ancestor that is not shared with bryophytes. We have
corrected this sentence in the Introduction of the revised manuscript to accurately
reflect fern phylogeny. The revised statement is: “As the sister group to seed plants,
ferns share a common ancestor with them, and thus provide crucial insights into plant
evolution, particularly transitions in anatomy, life history, and reproductive strategies”.
This clarification ensures the background is phylogenetically accurate.

2-Line 130 “These results collectively underscore the high contiguity, consistency, and
completeness of our *B. insignis* assembly.” I am concerned about the authors’
conclusion of completeness given the BUSCO score of 81.5% complete recovery (77.7%
based on annotation assessment, Supplementary Table 3). Despite having high LAI and
CRAN metrics, this is a very low BUSCO score (The expected standard for the field is
usually 95% complete or higher). At minimum, the authors need to address the possible

reasons for the low BUSCO recovery and discuss whether it is an artifact of their
assembly or possibly reflecting a biological reality (unlikely).

**Response:** Thank you for this important comment. We apologize for the overstatement
regarding assembly “completeness.” In the original manuscript, our BUSCO analysis
reported approximately 81.5% completeness for the genome assembly (and about 77.7%
for the annotated gene set), which is indeed lower than the typical >90–95% seen in
many seed plant genomes. We agree that we needed to provide an explanation and
adjust our claims accordingly.

We believe this is due to the complexity, large size, and high repetition of fern genomes,
along with the lack of a high-quality fern reference genome. As mentioned in our
response to Reviewer #1, the use of the embryophyta_odb10 dataset, which is biased
toward angiosperms, may not have been optimal for our study. In the revised version,
we repeated the BUSCO analysis with the updated viridiplantae_odb12 dataset
(updated July 1, 2025), which better represents non-seed plants. This more
comprehensive BUSCO set greatly improved our completeness estimates. The genome
assembly now scores 97.4% Complete, and the gene annotation scores 84.2%, as shown
in the updated Supplementary Table 1 (and Response Table 1). The large jump in the
assembly’s BUSCO score (from ~81% to ~97%) indicates that the prior lower score
was largely an artifact of using an angiosperm-skewed BUSCO dataset
(embryophyta_odb10) on a fern genome. In other words, many of the “missing”
BUSCO genes were likely present in *B. insignis* but not detected due to sequence
divergence or absence from the embryophyta BUSCO list.

**Response Table 1 | Updated BUSCO completeness assessment using the**
**viridiplantae_odb12 dataset**

Evaluation Type	Complete (C)	Single-copy (S)	Duplicated (D)	Fragmented (F)	Missing (M)	Total BUSCOs (n)
Genome Assembly	97.4%	89.9%	7.5%	2.1%	0.5%	822
Gene Annotation	84.2%	79.1%	5.1%	12.7%	3.2%	822

3-Line 187 “relative rate tests suggest that *B. insignis* has the slowest evolutionary rate
among core leptosporangiate ferns (Supplementary Table 7).” Given the information in
Supplementary Table 7, this is an overexaggeration. It appears that *B. insignis* does not
have a significantly slower rate of evolution (in 31 single copy genes) relative to several
other core leptosporangiate ferns, including *L. chinensis* and *D. repens*. Please amend
this statement and temper your conclusion that “this slow evolutionary rate may have
bolstered genomic stability, enabling...”.

**Response:** We sincerely appreciate the reviewer’s careful examination of our
statements on substitution rates. This comment echoes the concern raised by Reviewer
#1, and we have taken it very seriously. We have amended the manuscript to remove
the claim that *B. insignis* has the single “slowest” rate among core leptosporangiate
ferns and to significantly soften the interpretation about what a slower rate implies for
the species.

As detailed in our response to Reviewer #1 (Response Lines 61–92 above), we
recognize that our original wording was too strong and could be misleading. We have
rewritten the relevant passage in the Results/Discussion to clarify that while *B. insignis*
shows generally slower substitution rates compared to many ferns, it is not an outlier in
all comparisons. For instance, as you pointed out, species like *Lygodium chinensis* and
*Diplazium repens* have evolutionary rates statistically indistinguishable from *B.*
*insignis* in our single-copy gene analyses.

Responses to Reviewers' comments

Reviewer #1 (Remarks to the Author):

Line 184-186. First, I won't consider *Pronephrium simplex* a "close relative". It's in a different family. Second, I still don't understand why only single copy genes show rate difference, and not the low copy genes. Importantly, since single copy genes only account for a small proportion of the genome, can they really represent the evolutionary rate of whole genomes?

Response: We thank the reviewer for the taxonomic clarification and agree that *Pronephrium simplex* is not a close relative of *Brainea insignis*, we have therefore removed this phrasing from the manuscript.

With respect to the use of only single-copy genes, we acknowledge that they represent only a small fraction of the genome and thus may not fully capture genome-wide substitution rates. Nevertheless, analysis of low-copy or multi-copy genes are more vulnerable to confounding by complex gene family dynamics, including duplication and loss, sub- or neofunctionalization, and episodes of positive selection in individual paralogs. These processes introduce rate heterogeneity that can obscure lineage-level signals. For this reason, strict single-copy orthologs remain the standard approach for phylogenomic and molecular rate comparisons, as they minimize the risk of paralogs and other biases.

We also recognize the limitations of this approach, particularly given the large evolutionary distances among available fern genomes, and the fact that most taxa are represented by transcriptomes rather than high-quality whole genomes. To provide an independent line of evidence, we therefore conducted a genome-wide synteny analysis with the tree fern *Alsophila spinulosa*, a lineage previously reported to evolve slowly. Despite divergence over 200 Mya (TimeTree5), we detected extensive 1:1 syntenic blocks between *B. insignis* and *A. spinulosa* (Supplementary Fig. 6). This level of conserved collinearity supports our interpretation that *B. insignis* genome has evolved at a relatively slow pace, at least in terms of genomic architecture. This result has been added in the revised manuscript (Lines 189-193) and is presented in the supplementary material.

Together, these complementary analyses strengthen the inference of slow genome evolution in *B. insignis* while underscoring the need for additional high-quality fern genomes to enable broader and more precise assessments of evolutionary rate variation.

Inter-genomic comparison: *Alsophila spinulosa* vs *Brainea insignis* (9,553 gene pairs)

Supplementary Fig. 6 | Synteny analysis between *Alsophila spinulosa* and *Brainea insignis*.

Line 189-192. The relative rate tests similarly have issues. I am not sure to what extent 31 single copy genes are representative of this species. Further, with *Psilotum nudum* as the outgroup, the timescale here is vast. In other words, is the rate difference truly specific to *Brainea* or the hundreds of million years of evolution leading to it? I am not certain the analyses here are valid.

Response: We appreciate the reviewer's valuable comments. We agree that a set of 31 single-copy genes is limited and, on its own, cannot decisively represent genome-wide substitution rates. Accordingly, we have supplemented our study with results from genome synteny analyses, which clearly show that, compared with the more distantly diverged *Alsophila*, *Brainea* has retained a substantial amount of syntenic blocks (Supplementary Fig. 6). These findings provide structural evidence that *Brainea* has indeed evolved at a relatively slower pace, at least in terms of preserving genomic

architecture.

Regarding the relative-rate test, we applied the classical pairwise approach of Tajima, which compares the substitution counts between ingroup taxa relative to an outgroup O to test whether A–O and B–O distances differ significantly. We agree that relying solely on a distantly related outgroup such as *Psilotum nudum* may introduce artifacts due to substitution saturation over deep timescales. To mitigate this, we repeated all tests with two additional outgroups spanning shallower and intermediate divergence—*Marsilea vestita* and *Asplenium formosae*. As summarized in Response Table 1, the majority of significant comparisons consistently indicate a slower rate in *B. insignis* across outgroups. Concordance of the signal across multiple outgroup depths increases confidence that the effect reflects the *Brainea* lineage rather than deep stem accumulation.

Taken together—(i) the outgroup-robust relative-rate results and (ii) the independent evidence of conserved large-scale collinearity—our inference is that *B. insignis* has evolved comparatively slowly, at least in terms of genomic architecture, while we explicitly acknowledge the limits imposed by current fern genomic resources and the small single-copy gene set. Further high-quality genomes from closely related lineages will enable more precise, genome-wide rate estimation.

Response Table 1 | Relative rate tests comparing *B. insignis* with other ferns using *M. vestita* and *A. formosae* as outgroups separately

Ingroup1	Ingroup2	Outgroup	Genes	Identical	Divergent	Ingroup1 specific	Ingroup2 specific	Outgroup specific	Slow	CHI ² test statistic	P-value
Alsophila spinulosa	B. insignis	M. vestita	31	9261	694	567	628	1310	-	3.11	0.07763
Adiantum capillus-veneris	B. insignis	M. vestita	31	9426	714	602	457	1499	B. insignis	19.85	0.00001
Asplenium formosae	B. insignis	M. vestita	31	9273	558	488	351	1719	B. insignis	22.37	0.000001
Woodwardia prolifera	B. insignis	M. vestita	31	10017	231	143	189	2361	W. prolifera	6.37	0.01158
Pronephrum simplex	B. insignis	M. vestita	31	9773	313	241	246	2174	-	0.72	0.588
Bolbitis hekouensis	B. insignis	M. vestita	31	8905	429	304	271	1832	B. insignis	1.89	0.16876
Davallia repens	B. insignis	M. vestita	31	8993	496	377	314	1754	B. insignis	5.74	0.01655
Loxogramme chinensis	B. insignis	M. vestita	31	7990	511	441	309	1473	B. insignis	23.23	0.000001
Nephrolepis cordifolia	B. insignis	M. vestita	31	9282	429	268	302	1845	-	2.03	0.15442
Ingroup1	Ingroup2	Outgroup	Genes	Identical	Divergent	Ingroup1 specific	Ingroup2 specific	Outgroup specific	Slow	CHI ² test statistic	P-value
Woodwardia prolifera	B. insignis	A. formosae	31	10968	124	197	242	1079	W. prolifera	4.61	0.03173
Pronephrum simplex	B. insignis	A. formosae	31	10703	166	325	338	917	-	1.31	0.28
Bolbitis hekouensis	B. insignis	A. formosae	31	9987	243	436	357	790	B. insignis	7.87	0.00503
Davallia repens	B. insignis	A. formosae	31	10125	273	565	376	759	B. insignis	37.96	0.000001
Loxogramme chinensis	B. insignis	A. formosae	31	8929	274	662	354	658	B. insignis	93.37	0.000001
Nephrolepis cordifolia	B. insignis	A. formosae	31	10454	234	422	364	823	B. insignis	4.28	0.03857

Line 200-201. “Larger fern genomes tend to harbor a higher proportion of LTR-RTs with earlier insertion time”. How can you accurately estimate and compare insertion times when there is rate heterogeneity among fern lineages (a point you are trying to make above)?

Response: We agree that lineage-specific rate heterogeneity complicates cross-species comparisons of LTR-RT insertion ages, which are typically inferred from divergence between the 5' and 3' LTRs.

To address this variation, we adopted a phylogenetically informed, relative calibration

strategy. Specifically, we first estimated synonymous substitution rates (Ks) from orthologous genes across fern lineages, using fossil-calibrated divergence times from the TimeTree5 database to derive lineage-specific substitution rates ($r = Ks_{\text{ortholog}} / 2t$). These rates were then used to rescale observed Ks values, making them comparable with the panome Ks distribution of the focal species (via wgd-v2, developed by our collaborator Yves Van de Peer's group). This normalization corrects for background mutation rate differences, enabling cross-species comparisons of Ks distributions and alignment with shared evolutionary events such as WGDs. For LTR retrotransposons, we used LTR_retriever to calculate Ks between the 5' and 3' LTRs of each element. Rather than interpreting raw LTR-Ks values directly, we focused on relative patterns within the rescaled ortholog-based framework, which allowed us to assess shifts in the mode or peak of LTR-Ks distributions in a phylogenetically consistent manner. The observation that species with larger genomes retain not only more LTR-RTs but also proportionally older insertions—even after correction—supports our conclusion that genome size expansion is linked to the persistence of ancient LTR-RTs. While we recognize this represents a first-order correction and some residual rate variation may remain, the consistency of the results across lineages gives us confidence in the robustness and biological significance of this signal.

In summary, while absolute insertion times should be interpreted cautiously, our use of time-calibrated orthologous divergence to normalize rates provides a phylogenetically justified framework for comparing the relative ages of LTR-RT bursts across ferns. To enhance transparency and reproducibility, we added a detailed description of this rate normalization procedure in the Methods section (Lines 494-498).

Line 213-216. The vast majority of the genes are evolving under purifying selection ($\omega < 1$) and I do not believe this can be used to support “strong functional constraint on these lignin-related genes across these fern lineages.” In addition, *Brainea* and *Sphaeropteris* span deep evolutionary time, during which arborescence only accounts for a small blip in time – does this calculation actually mean anything?

Response: We thank the reviewer for raising this important point. We agree that $\omega < 1$ is a general pattern for most protein-coding genes, and we also acknowledge that the divergence between *Brainea* and *Sphaeropteris* spans a deep evolutionary timescale, during which the arborescent growth form represents only a relatively small fraction.

To further clarify, we extracted the lignin-related genes identified in *B. insignis* and examined their micro-synteny blocks with *Sphaeropteris lepifera* and *Alsophila spinulosa*. The results (presented in Supplementary Fig. 10) show that a subset of lignin-related genes is highly conserved across these species. Thus, our intention is not to imply a direct link between ω ratios and arborescence, but rather to highlight the

conserved nature of lignin pathway genes, consistent with their central role in cell-wall biology across ferns. We have therefore softened our original interpretation and no longer present ω values as primary evidence. Instead, we highlight an independent line of support: conserved microsynteny among a subset of lignin-pathway genes in *Brainea insignis*, *Sphaeropteris lepifera*, and *Alsophila spinulosa* (Supplementary Fig. 10). The long-term preservation of gene order and content across these deeply divergent lineages is more consistent with functional and dosage constraints acting on pathway architecture than with ω values alone, which can be ambiguous or misleading.

Supplementary Fig. 10 | Microsynteny analysis of lignin-related genes across three fern species

Line 219-222. Another possibility is that the expansion of these gene families has nothing to do with woodiness!

Response: We agree that CAFÉ-inferred expansions do not, by themselves, establish any causal relationship with woodiness. CAFE detects net turnover in family size along branches under a birth–death model; it is agnostic to function and trait causation. To avoid over-interpretation, we have removed the speculative causal statement and now present these results as descriptive patterns of gene-family turnover. We explicitly note that linking expansions to woodiness would require additional evidence—e.g., replicated associations across independent transitions using phylogenetic comparative models that control for genome size/WGD and life history, together with tissue-specific expression and functional assays. The manuscript has been revised accordingly.

Line 225-228. What’s the evidence for “convergent evolution of lignin specialization”? Do we know what types of lignin does *Brainea* actually have? There are a lot of vague associations and assumptions in this section.

Response: We thank the reviewer for this valuable comment. We agree that our original phrase “convergent evolution of lignin specialization” exceeded the available evidence. Indeed, the exact lignin composition of *B. insignis* has not been characterized, and thus direct support for convergent evolution is lacking. Our initial association between *Brainea* and *Stenochlaena palustris* was motivated by their shared family affiliation,

which led us to consider potential similarities in lignin-related traits. To avoid over-interpretation, we have removed this speculative conclusion and revised the text to present the interpretation more cautiously (Lines 224-229).

Minor comments:

Line 60. “Ferns share a common ancestor with them” – Every organism on this planet shares a common ancestor with another, so this statement is not needed.

Response: Thank you for pointing this out. We acknowledge that the statement could be seen as trivial given the universal common ancestry of life. It has been removed to improve precision.

Line 89. Who is “them” in “the broader biodiversity that depends on them”?

Response: We appreciate the reviewer’s meticulous comment. We have revised this sentence to: Given the species’ dual importance to conservation and evolutionary studies, understanding its genomic makeup and evolutionary resilience is critical for guiding the conservation of both the species itself and the biodiverse habitats it sustains.

Line 127. “Clipping-based Revealing Assembly Quality” should be “Clipping Reveals Assembly Quality” or “Clipping information for Revealing Assembly Quality”?

Response: We thank the reviewer for this suggestion. The term has been revised to “Clipping Information for Revealing Assembly Quality (CRAQ)” for improved clarity and accuracy.

Reviewer #2 (Remarks to the Author):

Dear Editors and Authors,

It is my opinion that Li et al. have satisfactorily addressed both reviewers’ comments in a manner that improves the execution of the study and better highlights its immediate and broader impacts. The methodologies are sound, of the standards of the field, reproducible, and noteworthy. As described in my earlier review, I believe this manuscript is a well-written and valuable contribution as the first complete Eupolyplod II fern genome assembly and as a rare example of using whole genome sequencing to investigate population level diversity for a fern species. I especially appreciate that the authors have broadened their discussion of the importance of this genome for investigating fern genome evolution. Of particular note, is their result that large genome size in the polypods seems largely dictated by TE expansion, rather than by WGD followed by slow rates of diploidization – this is important because it counters the

existing paradigm for how ferns have accrued large chromosome numbers and genome sizes relative to angiosperms. I believe this study will also serve as a useful and well-cited model for population genomics approaches in ferns, a group historically underrepresented in population genomics because of limited genomic resources and large population size.

One caveat is that I wish the manuscript had been assigned to a reviewer with expertise in the evaluation of genetic offset under future climate scenarios. I feel that this aspect of the manuscript largely went unevaluated and might benefit from an expert's opinion.

Response: We thank the reviewer for the thoughtful evaluation and endorsement. We have clarified how the *B. insignis* genome informs fern genome evolution (e.g., WGD context, TE dynamics) and explicitly noted that, to our knowledge, this is the first application of a genetic-offset framework in a fern. We also acknowledge the reviewer's caveat and, in the Discussion/Methods, more clearly justify the suitability and limitations of genetic-offset analyses for *Brainea* (long-lived, perennial, spore-dispersed) and the steps we took to mitigate confounding (see also responses to Reviewer #3, point 5).

Reviewer #3 (Remarks to the Author):

Xia et al. present a study that assembled a de novo genome of a cycad fern and conducted population genomic analyses based on 94 individuals. Although the dataset is substantial, primarily due to the species' large genome size, I find the novelty of the analysis somewhat underwhelming. Below, I outline several concerns and suggestions that could enhance the study's relevance and originality, particularly in comparison to other population genomic studies on species with smaller genome sizes.

Response: We appreciate the reviewer's careful assessment and constructive suggestions. Below we describe concrete changes and clarifications made to enhance originality, interpretability, and rigor.

1. Importance of the Genome and Comparative Analysis: While the de novo genome assembly of this endangered species within the Eupolypods II clade is significant, the genomic analyses presented in the study feel overly general. Beyond the basic genomic structure in Figure 1, Figure 2 highlights WGD (whole-genome duplication) events and phylogenetic relationships across diverse species. It would be more impactful to compare this genome with other published fern genomes to examine genome collinearity and divergence among closely related species. A more detailed analysis could investigate unique transposable elements (TEs) or gene expansions/contractions specific to this species, as well as large-scale structural variations in the genome.

Linking these variations to associated genes and their functional enrichment would provide deeper insight into this species' unique genomic features.

Response: We appreciate the reviewer's insightful suggestions. We agree that comparative genomic analyses, such as collinearity assessments, investigation of species-specific transposable elements, gene family dynamics, and structural variation, provide the most incisive route to uncover lineage-specific features. The current constraint is data availability: chromosome-level fern genomes remain scarce, and within eupolypods II our *B. insignis* assembly is, to our knowledge, the first at this resolution. The nearest available references (*Ceratopteris richardii*, *Adiantum capillus-veneris*) diverged from *Brainea* over 100 million years ago and exhibit substantial differences in chromosome number, reflecting deep karyotypic reorganization. Coupled with extensive gene turnover, large-scale rearrangements, and TE flux, this severely limits the resolution of fine-scale "one-to-one" comparative analyses. To avoid over-interpreting differences driven mainly by deep evolutionary divergence rather than true lineage-specific changes, we interpreted our comparative analyses with caution. We agree that as more high-quality fern genomes become available, broader and more detailed comparative studies—including genome size evolution, TE dynamics, synteny, and gene family turnover across ferns—will be both possible and highly informative.

In the revised manuscript, we therefore performed genome-wide collinearity analysis between *B. insignis* and *Alsophila spinulosa* (diverged >200 Mya) and observed extensive 1:1 syntenic blocks (Supplementary Fig. 6), highlighting the structural conservation of the *B. insignis* genome. This analysis also revealed a shared WGD event, but indicated that WGD alone cannot fully account for genome size variation in ferns; rather, LTR insertions and accumulation appear to play a more dominant role. Furthermore, micro-synteny analysis revealed that lignin biosynthesis-related gene sets are conserved between *B. insignis*, *Sphaeropteris lepifera*, and *Alsophila spinulosa* (Supplementary Fig. 10), despite their divergence over 200 Mya, underscoring the long-term evolutionary stability of these functionally critical pathways. Taken together, these findings provide important insights into fern genome evolution. We hope the reviewer appreciates the current limitations in fern genomic resources and recognizes that, although some analyses are standard in seed plants or animals, our study provides pioneering insights into fern genomics and population genetics.

2. Integration of Demographic, Divergence, and Gene Flow Models: The study constructs demographic, differentiation, and gene flow models separately. However, these processes are interrelated and should ideally be integrated into a cohesive framework. Typically, divergence models incorporate demographic and gene flow histories to provide more accurate and comprehensive inferences. Analyzing these

processes separately can lead to confounding effects and compromise the accuracy of the results. I suggest the authors revisit their fastsimcoal analysis, integrating demographic and gene flow scenarios into unified models for better reliability and interpretability.

Response: We thank the reviewer for this important suggestion. We agree that divergence, demography, and gene flow are inherently interdependent and ideally should be inferred within a unified modeling framework. While fastsimcoal2 does support such joint inference, simultaneous estimation of many parameters from the site frequency spectrum (SFS) can lead to challenges in parameter identifiability—such as trade-offs between divergence time, effective population size changes, and migration rates—as well as flat likelihood surfaces, which may compromise the reliability of estimates. To balance realism with statistical robustness, we adopted a stepwise, hierarchical modeling strategy: we first fitted simpler models with fewer parameters (e.g., divergence without migration), then incrementally introduced additional processes (asymmetric continuous migration and pulse admixture), retaining complexity only when supported by the data. Practically, each step was evaluated by repeated optimizations from diverse starting points, and model choice was based on Akaike Information Criterion (AIC) and Δ AIC. The best-supported models therefore integrate divergence, demographic history, and gene flow rather than treating them in isolation, and their parameter estimates are reported accordingly (Supplementary Figs. 17–19; Tables 8–10). Where appropriate, we also compared key time-scale parameters with independent skyline-based inferences to assess concordance.

Such a tiered modeling framework has also been applied in recent studies—for example, Xiao et al., (2023, PNAS) used a similar progressive approach to robustly reconstruct the domestication history of grapevine. We have revised the text to make this workflow explicit and to avoid the impression that processes were analyzed independently. The stepwise nesting serves only to ensure parameter identifiability; the final inferences are drawn from the highest-supported integrated models (Lines 557–566).

Reference: Xiao et al., Adaptive and maladaptive introgression in grapevine domestication. *Proc. Natl. Acad. Sci.* 24 (120), 2023.

3. Genetic Load and Population Decline: One of the study's key contributions is its emphasis on the endangered status of this species and the impact of population decline on genetic variation and genetic load. However, the current analysis only compares genetic load across the three genetic groups, which exhibit minimal differences and provide limited insight into the extent of genetic load accumulation. To enhance this analysis, the authors could employ established methods for measuring genetic load, such as the 0-fold/4-fold ratio or the homozygous/heterozygous counts of loss-of-

functon(LoF) and deleterious mutations. Furthermore, comparing the genetic load of this species to data from other studies on species with similarly small population sizes or those in an endangered state would offer a more meaningful context. This comparison could help clarify the severity of the genetic load accumulation in this fern species and its relative position on the spectrum of endangerment.

Response: We agree that assessing genetic load with site-class-based metrics and homozygosity of deleterious variants provide a clearer picture than groupwise comparisons alone. Accordingly, we expanded the analysis in three ways. First, we derived 0-fold and 4-fold degenerate sites from the reference CDS and computed the ratio of nonsynonymous to synonymous diversity (π_0/π_4) to gauge the efficacy of purifying selection (Supplementary Table 11). Second, we quantified the burden of deleterious alleles by distinguishing homozygous and heterozygous genotypes for predicted loss-of-function (LoF) and deleterious variants, and we summarized the proportion of homozygous deleterious genotypes per individual as a proxy for realized load (Fig. 4e–f). Third, we interpreted these patterns in the context of recent population decline, where increased drift and inbreeding are expected to elevate realized (homozygous) load even if the total number of deleterious alleles remains similar across groups.

For cross-species context, resources in ferns are still limited; to our knowledge, *Alsophila spinulosa* is the only species with both a high-quality genome and population resequencing suitable for comparison. We therefore discuss *Brainea insignis* relative to *A. spinulosa* in the manuscript (Lines 327–337), noting life-history and reproductive differences that preclude direct quantitative benchmarking against seed plants. This provides the most biologically relevant frame of reference currently available while avoiding potentially misleading comparisons across very different clades.

Together, these additions move beyond coarse group comparisons to a site-class and genotype-state perspective, offering a more informative view of genetic-load accumulation associated with recent demographic collapse in *B. insignis*, while clearly stating the comparative limits imposed by current fern genomic resources.

4. Issues with the Ratio of Nonsynonymous to Synonymous Diversity: On line 312, the authors report a nonsynonymous-to-synonymous diversity ratio close to 1, which suggests a potential calculation error. I recommend carefully reviewing the methodology used to compute this ratio. If the result is accurate, it represents an unusual or unique finding, as such a high value (π_0 -fold/ π_4 -fold) is rarely observed. In that case, further investigation is necessary to determine the underlying causes of this pattern.

Response: We thank the reviewer for identifying this issue. The initially reported π_0/π_4 value was inflated due to a mistake in the definition of degenerate sites: non-coding

regions were inadvertently included in the annotation of 0-fold and 4-fold degenerate sites, which should be restricted to coding sequences (CDS). This contamination compromised the accuracy of the ratio, rendering it unreliable. We have corrected the analysis using properly filtered sites, and all results have been updated accordingly (Supplementary Table 11).

We re-ran the analysis using a codon-aware, population-aware pipeline consistent with recent practice. Specifically, we (i) annotated true 0-fold and 4-fold sites from the reference CDS using a degenerate-site annotator (i.e., *degenotate*; Mirchandani et al., 2024), (ii) intersected these coordinates with the population VCF to retain only polymorphisms at those sites, and (iii) recomputed nucleotide diversity (π) with ANGSD. The corrected π_0/π_4 estimates are substantially lower than the original near-unity value (see Supplementary Table 11), confirming that the initial result was an artifact of site misclassification.

Interpreted biologically, the corrected ratios remain elevated relative to many published plant datasets (e.g., Plomion et al., 2018), which is consistent with relaxed purifying selection and increased drift/inbreeding associated with recent population decline. We have revised the Results and Methods accordingly (Lines 314–316; 587–592) and are grateful to the reviewer for prompting this correction.

Reference:

[1] Mirchandani et al., 2024. High-throughput variant calling workflow for population genomics. *Molecular Biology and Evolution* 41(1).

[2] Plomion et al., 2018. Oak genome reveals facets of long lifespan. *Nature Plants* 4, 440–452.

5. Genetic Vulnerability to Climate Change: The study includes an analysis of genetic vulnerability to future climate change, but this is now a fairly standard approach. It's important to critically assess whether such methods are suitable for this species and whether they provide accurate predictions. The genetic offset analysis assumes environmental adaptation, but in this case, geographic distance is strongly correlated with environmental distance. This makes it difficult to disentangle the effects of selection-driven environmental adaptation from neutral genetic drift due to geographic separation. To address this issue, the authors could focus on specific identified adaptive variants and test whether they are associated with environmental variables, geographic differences, or a combination of both. Differentiating these factors is essential for understanding the role of environmental adaptation before conducting genetic offset analyses.

Response: We thank the reviewer for this thoughtful suggestion. We agree that genetic

vulnerability analyses are now a widely applied and relatively standard approach. We considered the approach suitable for *B. insignis*, a perennial, tree-like fern, and to our knowledge this represents the first application of this approach in ferns.

To guard against IBD confounding and to ensure that offsets reflect putative environmental adaptation, we implemented a two-step strategy and have clarified these details in the revision (Lines 614–617). First, we partitioned variants into neutral and adaptive sets. Putatively adaptive SNPs were identified using structure-aware scans (LFMM) and redundancy analysis (RDA) with climatic predictors; both methods account for background structure, and we retained only SNPs supported across approaches. We then evaluated isolation by environment (IBE) versus isolation by distance (IBD) using Mantel and partial Mantel tests separately for neutral and adaptive sets. The adaptive set retained strong environment–genotype associations after controlling for geographic distance, supporting a role for environment-linked selection beyond spatial drift. Second, we computed genetic offset exclusively on the adaptive SNP set, thereby focusing the analysis on loci most plausibly involved in environmental response. We interpret the resulting offsets as relative genomic vulnerability under the climate scenarios considered and explicitly note caveats relevant to *B. insignis* (e.g., potential microclimatic buffering, limits to long-distance spore dispersal, and unmodeled genotype-by-environment interactions).

These clarifications make explicit that (i) the offset analysis is conditioned on evidence for environment-associated adaptation and (ii) the conclusions are framed as comparative risk rather than precise prediction. We thank the reviewer for prompting these improvements in rigor and transparency.

6. Quantification of Genetic Offset Results: The genetic offset analysis currently presents results in the form of a colored plot, which makes it difficult to pinpoint which populations are more vulnerable. I recommend quantifying the offset values (e.g., local, forward, and reverse offsets) for each population or genetic cluster and visualizing them using boxplots or similar approaches. This would allow readers to clearly identify which populations are better adapted and which are more vulnerable to future climate change.

Response: We agree that integrating local, forward, and reverse offsets into a single multicolored plot may obscure some of the underlying information. In the revision, we therefore quantified genetic offset at the population level for each metric—local, forward, and reverse—and visualized them separately, which are provided in the Supplementary Materials (Supplementary Figs. 25-26). The corresponding text has been revised in the manuscript (Lines 372-390). This revision provides a more complete picture and allows readers to more clearly identify which populations are relatively well

adapted and which may be more vulnerable under future climate scenarios. We sincerely appreciate this experienced recommendation, which has improved the clarity and interpretability of our results.

Minor Comments

On line 241, the term “core variants” is unclear. How is it defined? Providing a clear definition would improve clarity.

Response: We thank the reviewer for pointing this out. As shown in Supplementary Fig. 12, “core variants” refers to the initial set of high-quality SNPs obtained after stringent filtering. All downstream datasets (Datasets1–3) were derived from this filtered SNP set. The term was used purely as a technical label without any special biological meaning. To avoid confusion, we have now clarified this definition in the Methods section (Lines 535-537).

In general, the Results section contains several ambiguous statements, partly because the methods are described later. Restructuring or clarifying these sections would make the study easier to follow.

Response: We appreciate this helpful comment. We have revised the Results to eliminate ambiguity arising from deferred methods. Specifically, we (i) define technical terms at first mention and provide brief one-sentence method primers, (ii) add explicit forward references to the corresponding Methods subsections, and (iii) align the order of Results subsections with the analysis workflow. For example, when introducing the high-confidence SNP set (formerly “core variants”; Lines 240–245), we now define it in situ and point to details described in the Methods. Throughout, figure/table callouts are synchronized with the text and acronyms are expanded at first use. These changes improve readability without duplicating Methods content.

Responses to Reviewers' comments

Reviewer #1 (Remarks to the Author):

I believe the authors have nicely addressed most of my previous concerns. I found the new synteny comparison with *Alsophila* very interesting. I do want to point out that computing “Ks” between LTR repeats is not valid. Ks is synonymous substitution and would only apply to protein-coding genes. How can you calculate Ks for repeats?

Response: We appreciate the reviewer's careful reading. You are absolutely right: Ks (synonymous substitutions per synonymous site) applies only to protein-coding genes. In our study, we used LTR_retriever, which applies the Jukes-Cantor (JC69) model to estimate the nucleotide substitution rate for noncoding sequences, thereby inferring LTR insertion times. The corresponding JC69 formula is $K = -\frac{3}{4} \ln(1 - \frac{4D}{3})$, where K is the corrected nucleotide substitution rate rather than Ks. We inadvertently referred to this as “Ks” in the text. This change is strictly terminological that does not alter any results or interpretations, and we have revised the wording consistently throughout the Methods section (Lines 502-506).

Reviewer #3 (Remarks to the Author):

I appreciate the efforts of the authors in addressing my previous comments, most of which have been resolved. Thank you for your efforts. However, I have two minor comments below:

(1) Although the analysis of genetic load in ferns is limited, it would still be interesting to compare the genetic load levels with those of other plant species, such as angiosperms. I believe genetic load data are available for more than 20 species, and it would be valuable to compare the genetic load of endangered ferns with other endangered plant species across phylogenetic positions.

Response: We thank the reviewer for this valuable suggestion. Given that genetic background (e.g., differences in substitution rates), sequencing strategies, data processing pipelines, and methods for measuring genetic load can vary considerably among studies, adopting an appropriate and comparable approach is crucial for interspecific comparisons. To enable fair cross-species comparisons amid variations in pipelines and ascertainment biases, we employed the widely used proxy π_0/π_4 (nonsynonymous to synonymous diversity), which reflects the efficacy of purifying selection and correlates with genetic load.

We compiled π_0/π_4 ratios from >40 plant species spanning ferns and seed plants, and added a new Supplementary Figure summarizing these values (Supplementary Fig. 21).

The results show that the endangered ferns (*Brainea insignis* and *Alsophila* spp.) exhibit significantly elevated π_0/π_4 relative to most seed plants, with the exception of *Acer yangbiense*, a critically endangered species with extremely small effective population size (N_e). These patterns are consistent with the expectation that small effective population sizes weaken purifying selection, thereby increasing the burden of deleterious polymorphisms.

We acknowledge that future studies incorporating more genomic data may allow for further updates or refinements of these results. Nevertheless, we believe that presenting this comparative analysis in the supplementary materials provides valuable contextual information and serves as a useful reference for understanding the genetic load characteristics of endangered ferns relative to other plant groups.

Supplementary Fig. 21 | Summary of π_0/π_4 ratios across fern and seed plant species.

(2) The forward and reverse differences across different populations shown in Supplementary Figure 25 are difficult to distinguish, likely because the local offset values are too high. I suggest that instead of using the same value range for all the plots, each figure should use its own value range for plotting the offset. This adjustment would help to clearly visualize the landscape distribution of offset values and make it easier to differentiate between forward and reverse offsets.

Response: Thank you for the suggestion. We have adjusted the plotting ranges so that the local, forward, and reverse offset maps now each use their own respective value ranges, instead of the previous approach that applied a range based solely on the local offset values (Supplementary Figure 26). This adjustment has indeed significantly

enhanced the visual distinguishability between forward and reverse offsets, as illustrated below. Importantly, these changes do not affect the original results; they only improve the presentation of the offset landscapes.

Original Supplementary Fig. 25

Revised Supplementary Fig. 26

Revision #4
Responses to Reviewers' comments

Reviewer #3 (Remarks to the Author):

I appreciate the authors for addressing my concerns. I have no further comments and look forward to seeing this work published soon.

Response: We sincerely thank Reviewer #3 for the constructive feedback in the review process. Your comments have greatly improved our manuscript and aligned it better with the journal's standards. We appreciate your support.